# Controlling piezoresistance in single molecules through the isomerisation of bullvalenes

Jeffrey R. Reimers [1,2] ✉, Tiexin Li[3], André P. Birvé[4], Likun Yang[1], Albert C. Aragonès [5,6], Thomas Fallon [4] ✉, Daniel S. Kosov [7] ✉ & Nadim Darwish [3] ✉

Nanoscale electro-mechanical systems (NEMS) displaying piezoresistance offer unique measurement opportunities at the sub-cellular level, in detectors and sensors, and in emerging generations of integrated electronic devices. Here, we show a single-molecule NEMS piezoresistor that operates utilising constitutional and conformational isomerisation of individual diaryl-bullvalene molecules and can be switched at 850 Hz. Observations are made using scanning tunnelling microscopy break junction (STMBJ) techniques to characterise piezoresistance, combined with blinking (current-time) experiments that follow single-molecule reactions in real time. A kinetic Monte Carlo methodology (KMC) is developed to simulate isomerisation on the experimental timescale, parameterised using density-functional theory (DFT) combined with non-equilibrium Green's function (NEGF) calculations. Results indicate that piezoresistance is controlled by both constitutional and conformational isomerisation, occurring at rates that are either fast (equilibrium) or slow (non-equilibrium) compared to the experimental timescale. Two different types of STMBJ traces are observed, one typical of traditional experiments that are interpreted in terms of intramolecular isomerisation occurring on stable tipped-shaped metal-contact junctions, and another attributed to arise from junction–interface restructuring induced by bullvalene isomerisation.

It is now possible to observe single-molecule chemical reactions using scanning tunnelling microscopy (STM) beyond simple electron-transfer reactions[1–5]. Electrocatalysis can be used to drive reactions[6], as the electric field inside the STM can modify molecules[7] as well as to measure their properties[1,6,8–16], and electric-field driven redox and interfacial reactions have been shown to lead to unprecedented rectification in molecular devices[17,18]. In alternate technology, STM junctions have been modelled[19] and shown[20–22] to lead to interfacial structural changes through mechanical control of the junction separation. This is of practical significance as it can lead to piezoresistance in a nano electro-mechanical system (NEMS) measured on the Å atomic scale.

[1]International Centre for Quantum and Molecular Structures and the Department of Physics, Shanghai University, Shanghai 200444, China. [2]School of Mathematical and Physical Sciences, University of Technology Sydney, Sydney, NSW 2007, Australia. [3]School of Molecular and Life Sciences, Curtin University, Bentley, WA 6102, Australia. [4]School of Environmental and Life Sciences, University of Newcastle, Callaghan, NSW 2308, Australia. [5]Department of Materials Science and Physical Chemistry, University of Barcelona, Marti i Franquès 1, 08028 Barcelona, Catalonia, Spain. [6]Institute of Theoretical and Computational Chemistry, University of Barcelona, Diagonal 645, 08028 Barcelona, Catalonia, Spain. [7]College of Science and Engineering, James Cook University, Townsville, QLD 4811, Australia. ✉e-mail: jeffrey.reimers@uts.edu.au; thomas.fallon@newcastle.edu.au; daniel.kosov@jcu.edu.au; nadim.darwish@curtin.edu.au

Indeed, length changes can be detected by monitoring changes in the junction conductance associated with molecular stretching or junction rearrangement[20–35]. Herein, constitutional and conformational isomerism is combined and used to: (i) create fast-switching transistor-like circuits that can exploit single-molecule isomerisation on the ms timescale, (ii) detect oscillating reactions in junctions and use the resulting alternating-current profiles as indicators of junction properties, and (iii) reveal processes leading to junction formation that are critical[36] to basic understanding of molecular junctions.

First, piezoresistance is demonstrated that is stable over device extensions of up to 9 Å, through the use of bullvalene shape-changing[37,38] molecules (Fig. 1a) as the active molecular component. Bullvalenes can change shape by constitutional isomerism, exploiting a succession of sigmatropic Cope rearrangements[39] reactions, that occur[40] (at 25 °C in $CCl_2 = CCl_2$ solution) at a rate of 3.2 kHz, to make the structure fluxional. This constitutional isomerism, as well as configurational isomerism associated with the diaryl bullvalene linkages, could control conductance. To investigate this, electrical circuits are formed via the STM break junction (STMBJ) technique in which an Au STM tip is crashed into an Au(111) substrate and then retracted[41], allowing single molecules to bridge the broken contacts (Fig. 1b). This technique allows conductance to be measured as the electrode-to-electrode separation spanned by the bridging molecule is varied (Fig. 1d). Conduction peaks typical of STMBJ experiments and NEMS piezoresistance devices are observed following junction formation.

Then, the STMBJ blinking (current–time) approach[1,42,43] is used in which an STM tip is brought near the substrate, without making mechanical contact, and held at a fixed, known, separation. Single molecules become trapped between both STM electrodes, forming molecular junctions that remain for times up to ≈2 s. The blinking approach is utilised to track single-molecule isomerisation events in real time, establishing that isomerisation is responsible for the observed piezoresistance (Fig. 1b, c). This also facilitates means of measuring junction properties through the detection of isomerisation-controlled alternating currents.

To interpret these experiments, a kinetic Monte Carlo (KMC) scheme is introduced for the modelling of isomeric composition on the experimental timescale, parameterised by density-functional theory (DFT) coupled with non-equilibrium Green's function (NEGF) conductance calculations[44]. In addition, DFT/NEGF molecular dynamics (MD) simulations of junction extension are performed. These simulations explore possible connections between piezoresistance and bullvalene isomerism, including: (i) constitutional isomerism, which changes through-bond conduction pathways and could also modulate quantum interference effects[45,46], (ii) bullvalene-linkage conformational isomerism, which could also modulate conductance[19,20,26,27,34,47,48] and related properties[49], (iii) π-stacking of the aryl groups controlled by both constitutional and conformational isomerism that can modulate conductance[32] and induce interference effects[28], and (iv) junction interface rearrangements[24].

An application of the NEMS technology is then developed, based on the observation that two types of conductance traces are observed in the STMBJ experiments. One type of trace presents results that cannot be interpreted using the atomic models that have been widely applied[15,16,50,51] for gold–molecule–gold junctions. These results are interpreted in terms of isomerisation-driven tip reconstruction (Fig. 1e), and are suggestive that adsorbate binding may, in general, be a significant process in determining the atomic nature of gold contacts in STMBJ experiments.

## Results and discussion
### Synthesis
The *bis*(4-thioanisole)bullvalene was prepared from *bis*(Bpin)bullvalene through a Suzuki cross-coupling reaction[38] with 4-bromothioanisole (Fig. 2a). A general overview is provided in methods, with full details and characterisation provided in Supplementary Information Note 1.

For a disubstituted bullvalene ensemble of this type, twelve constitutional isomers may in-principle form[52], named **A – L** in Fig. 2b, that also embody three sets of enantiomers (**B, B′**), (**D, D′**), and (**E, E′**). A network diagram of the ensemble is shown (Fig. 2c) whereby nodes represent isomers and edges represent Cope rearrangement pathways[39]. The solution phase populated isomer distribution was determined from low-temperature NMR spectra (see Supplementary NMR spectra) as isomers **A:B:C** in the ratio 53:33:13. Results from DFT calculations are consistent with this observation (see Supplementary Table 1).

### STMBJ single-molecule stretching experiments
Single-molecule conductance was measured using the STM in which an Au STM tip is pushed in and out at a rate of 0.5 Å ms⁻¹ from the surface in the presence of 50 μM solution of the bullvalene molecule in 1,2,4–trichlorobenzene (TCB), see Methods. Some current decays, depicting conductance in units of the "quantum of conductance" $G_0 = 2e^2/h = 77.5 \mu S$, as a function of time, are shown in Fig. 3a. These are representative of two trace (current decay) categories, evident from statistical analyses of 2425 current decays presented in Fig. 3b–d. Briefly, one shown type of decays depict conductance plateaus near 80–200 $\mu G_o$ lasting up to 10 ms, corresponding to tip retractions of 3–5 Å (red box), whereas the other types of decays show conductance steps that are stable for a short time in this region, but then switch to one, and then sometimes to another of much lower conductance level at which it also remains stable over distance of up to 3 Å (black box and blue box).

A statistical analysis of probability versus the logarithm of the conductance is shown in Fig. 3c. It shows a main conductance step feature (red arrow) centred at 100 $\mu G_o$ that ranges from 30 to 500 $\mu G_o$. A useful characterisation[48] is the ratio of the conductances at the high and low half maxima $\eta = HM_+ / HM_-$, which is 3.6. This is close to the

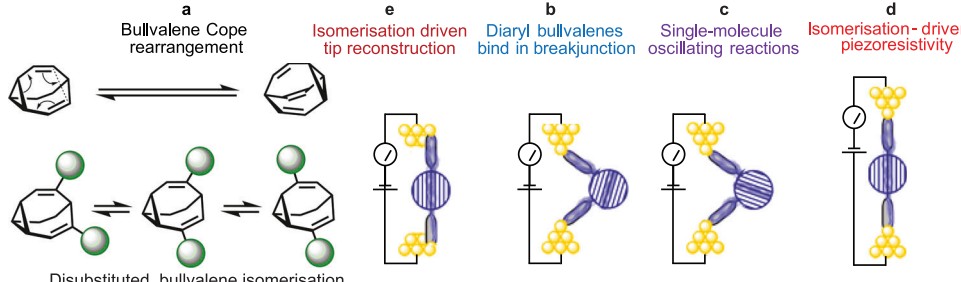

**Fig. 1 | Molecular and interfacial structural changes in solution and in NEMS devices. a** Sigmatropic Cope rearrangements make bullvalene a fluxional molecule in solution. **b** Diaryl substituted ($A_r$ = *para* ($C_6H_4$)–$SCH_3$) bullvalenes bind as bent isomers at short tip-tip distances in STMBJ experiments. **c** At specific tip extensions, bullvalene isomers with different conductances appear in equilibrium, allowing oscillating single-molecule reactions, occurring on the ms timescale, to be followed. **d** Tip retraction induces bullvalene isomerisation that controls conductance, manifesting piezoresistance. **e** Bullvalene isomerisation at short tip distances drives tip reconstruction. Blue colour in (**b–e**) represents possible electron pathways.

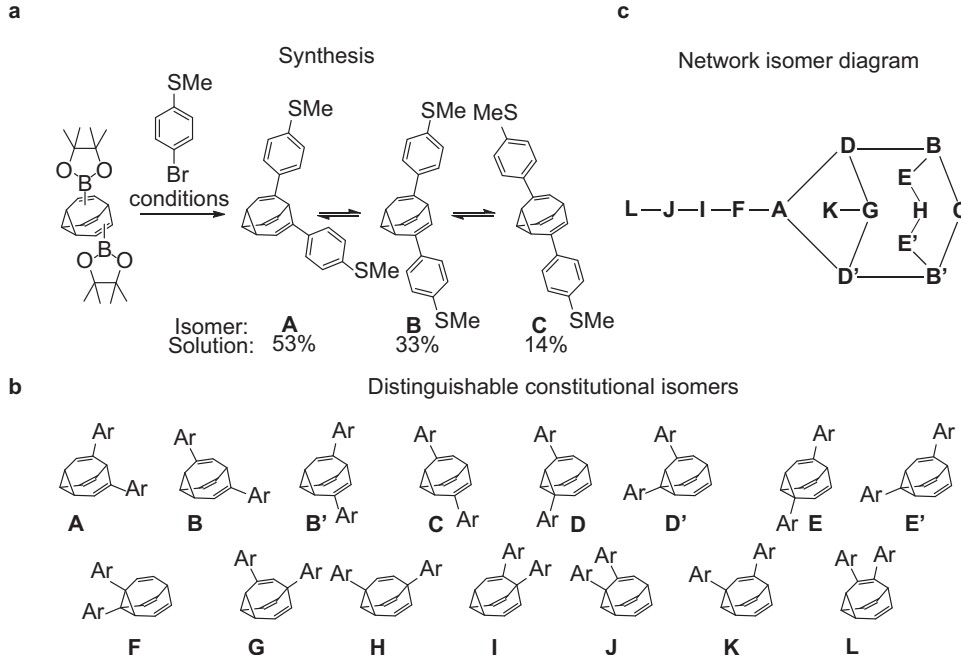

**Fig. 2 | *bis*(4-thioanisole)bullvalene isomerism and isomerisation. a** Synthetic conditions and observed isomers in solution: 4 bromothioanisole 2.2 eq., Pd₂(dba)₃ 20 mol%, [HP(t-Bu)₃]BF₄ 80 mol%, NaOH, THF/H₂O. 65 °C, 85% yield. **b** Possibly distinguishable constitutional isomers **A − L**, highlighting also 3 sets of enantiomeric pairs (**B, B'**), (**D, D'**), and (**E, E'**). **c** Network isomer diagram showing the possible Cope rearrangements.

value of 3.9 pertaining to chemisorbed alkanes HS-(CH₂)ₙ-SH observed[48] under similar experimental conditions. It reflects the typical spread of molecular conductances observed in STMBJ experiments and is traditionally attributed to variability on the gold–sulfur junction geometry. Much smaller values[48] of $\eta = 1.5$ are found in alternative STM blinking[53] experiments in which the conductance is measured on ordered self-assembled monolayers (SAMs) with regular interface geometries.

Uncharacteristic of STMBJ molecular conductance measurements is that a smaller population of the junctions/events form conductance peaks (9 and 13 $\mu G_o$) that are distinct from the main conductance peak at 100 $\mu G_o$, becoming more apparent as the conductance falls towards the noise limit of the current amplifier. Analogous results obtained using a non-isomerisable control molecule *para*-(4,4'-bi(methylthio) terphenyl instead of the bullvalene comprising the same S–CH₃ molecule–electrode contacts are provided in Supplementary Fig. 11 that does not manifest this feature. This suggests that the uncharacteristic features owe to bullvalene isomerisation. The most prominent unexpected peaks are highlighted by blue and black arrows in Fig. 3c; these are reproducible features, but they do show some bias dependence, see e.g., Supplementary Fig. 10. This could arise from the structural changes induced by the electric field. Most striking, however, is the narrow widths of all of these additional peaks, e.g., $\eta = 1.1$ (blue arrow) and 1.2 (black arrow). These values are less than that typically found for ordered molecular junctions in blinking experiments[48] and indicate that the interface geometries that facilitate conduction at low to moderate bias voltages are very regularly reproduced. Figure 3b shows that the 9 and 13 $\mu G_o$ conductance peaks in Fig. 3c arise as the tip is retracted to separations too large to support conductance via the mechanism operative for the much more prominent 100 $\mu G_o$ peak.

Correlation between the observed conductance peaks is revealed by the 2D correlation conductance map[54,55] shown in Fig. 3d. The diagonal line in this plot represents the same statistical information in Fig. 3c in another way, with the arrowed features from Fig. 3c labelled "1" (blue arrow), "2" (black arrow), and "3" (red arrow) on Fig. 3d. Away from the diagonal, the plot shows the probability that a single current trace

manifests two different conductance signatures within it, identifying regions labelled "4" (9 and 13 $\mu G_0$), "5" (9 and 100 $\mu G_0$), and "6" (13 and 100 $\mu G_0$). The weak correlations detected in regions 5 and 6 indicate that the traces only occasionally evolve from plateau regions at high conductance (100 $\mu G_0$) to plateaus at low conductance (9 and 13 $\mu G_0$). In contrast, the significant correlation in region "4" indicates that the two low conducting states often appear together, as some of the traces in Fig. 3a depict. That the observed correlations are weak between the high and low-conducting states, with the higher conducting signal dominating the overall counts of the histogram, also indicate that most conductance traces show junction breakage occurring from the high-conducting state before any transformation occurs to the two lower conducting states. Deconvolution of the peaks evident in Fig. 3c suggest that ≈7–9% of the conductance traces manifest transformations before junction breakage. The sharpness of the low conducting peaks ($\eta \approx 1$) can be explained by tip retraction process generating highly-ordered gold–molecule interfaces. In principle, changes in the junction structure and/or isomerisation processes within the bullvalene could account for these observed conductance changes.

## STM blinking experiments

In these experiments, a low solute concentration is used to prevent SAM formation on the Au substrate, and the STM tip is brought near the surface slowly, without contact, until a through-solution conductance is reached that is indicative of a desired substrate-tip separation $d$, see Methods and Supplementary Note 1. Occasionally, a bullvalene molecule then bridges the gap between the tip and the surface, and this leads to a sudden jump in the detected current (blink) owing to the charge transport through the connected molecule. Typically, these blinks last 0.2–2 s and are used to extract the temporal variation of conductance of a single-molecule junction in a controlled environment.

Some representative blinking (current–time) traces are provided in Fig. 4a, showing current as a function of time for electrode–electrode spacing set at $d = 11.8$, 12.8, and 16.2 Å, with conductance histograms built for 100 traces each reported in Fig. 4b. All three histograms

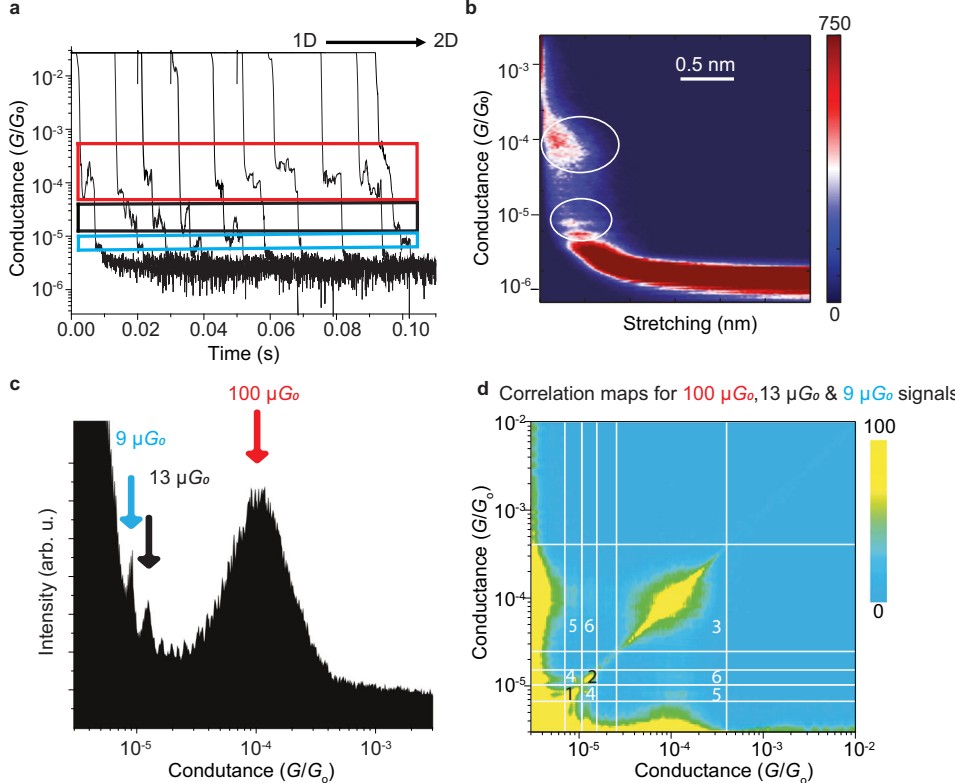

**Fig. 3 | STM measurements of the conductance of diaryl bullvalenes caught between two gold electrodes at a bias voltage of 50 mV in 1,2,4–tri-chlorobenzene solution using a tip retraction rate of 0.5 Å ms⁻¹ and sampling rate of 30 kHz. a** Representative conductance traces. Highlighted in red is single molecule plateaus near $100\,\mu G_0$, in black the plateaus at near $13\,\mu G_0$ and in blue the plateaus near $9\,\mu G_0$. **b** Two-dimensional conductance versus distance histogram of conductances accumulated from 2425 individual traces. The yield of the junctions, those which showed clear plateaus, is 48%; the two-dimensional conductance histograms of the entire 5067 curves collected are shown in the supporting information (Supplementary Fig. 10a). Circles highlight the features that correspond to the those highlighted by arrows in Fig. 3c. **c** One dimensional conductance histograms, with three conductance signals appearing as peaks near 100

$\mu G_0$ (red arrow, $\eta = 3.6$), $13\,\mu G_0$ (black arrow, $\eta = 1.2$), and $9\,\mu G_0$ (blue arrow, $\eta = 1.1$). **d** 2D correlation conductance map for the same data used to build the 1D conductance histogram in (**c**). Intersection regions labelled 1 to 6 correspond to the comparison of the relevant conductance signals at 100, 13, and $9\,\mu G_0$: regions 1–3 form the diagonal that reproduces the 1D-conductance histograms in (**c**), whilst off-diagonal regions 4–6 give the strength of the correlation between each two of the three different conductance states. The dark blue colour in (**b**) represents the absence of data (0 counts) while the red colour represents the maximum counts for accumulated data, as indicated in the colour-bar scale. The light-blue colour in (**d**) represents the lack of correlation between data while the yellow colour represents the highest correlation.

contain broad low-conductance features centred between 0.4 and 7 $\mu G_0$ associated with background conduction. They also show molecular conduction peaks, located at ≈150 $\mu G_0$. At the largest extension of 16.2 Å (blue trace), this peak become weak, however, with instead a strong peak appearing at a lower conductance of 8 $\mu G_0$.

In Fig. 4c, an individual conductance trace is shown that is representative of 10% of the total number observed. This trace starts at the background level, jumps to ≈8 $\mu G_0$, jumps again to ≈150 $\mu G_0$, and then falls to the background level. The middle jump is indicative of junction isomerisation, whereas the terminal changes are associated with junction formation and breakage. The switching between 8 and 150 $\mu G_0$ occurs at a rate of 2 h⁻¹ with an average duration of 0.7 s for the blinks at the 150 $\mu G_0$ level.

Further, the trace at 16.2 Å shown in Fig. 4d depicts continuous switching between two conductance states that differ in conductance by 2–5 fold. This trace is typical of 80% of the traces observed at that separation. Fast-Fourier transform of the oscillatory traces, shown in Fig. 4e, indicates a switching-rate profile that peaks at 850 Hz and varies between 750 and 1000 Hz. The oscillations are attributed to in-situ single-molecule bullvalene isomerisation (Supplementary Fig. 12). Results for analogous blinking experiments, performed using a non-isomerisable control molecule *para*-(4,4′-bi(methylthio)terphenyl with the same S–CH₃ molecule–electrode contacts, are provided in

Supplementary Fig. 13 and in Fig. 4f; for it, no analogous oscillations are observed.

## Modelling isomer properties in the gas phase

DFT calculations are performed for the structure and energetics of the constitutional isomers **A** – **L** in the gas phase, see Supplementary Table 1. As **A**, **B**, and **C** are observed in solution NMR, computations are used to investigate their conformational isomerism about the two bullvalene to aryl bonds. Stable conformers with torsional angles near ±40° are predicted, and for convenience these structures are named **A**ₘₘ, **A**ₘₚ, **A**ₚₘ, **B**ₘₘ, **B**ₘₚ, **B**ₚₘ, **B**ₚₚ, **C**ₘₘ, **C**ₘₚ, and **C**ₚₘ, where "**p**" and "**m**" indicated positive and negative torsional angles, respectively (Supplementary Table 1). Note that as **A** and **C** have planar symmetry, gas-phase conformers **A**ₘₘ and **A**ₚₚ are equivalent, etc.

The constitutional isomers depict very different relationships between the aryl substituents, with S – bullvalene centre – S bond angles ranging between 55° and 152°, whilst the S – S distances range from 7.0 to 15.0 Å (Supplementary Table 1). The equilibrium distances for the **A** isomers are ≈11.0 Å compared to ≈12.4 Å for **B** and **C**. Other isomers considered, with predicted gas-phase energies low enough to possibly form in STMBJ experiments, include: **D**, which is relevant as an intermediate in the reaction pathway linking **A** to **B** and henceforth **C**,

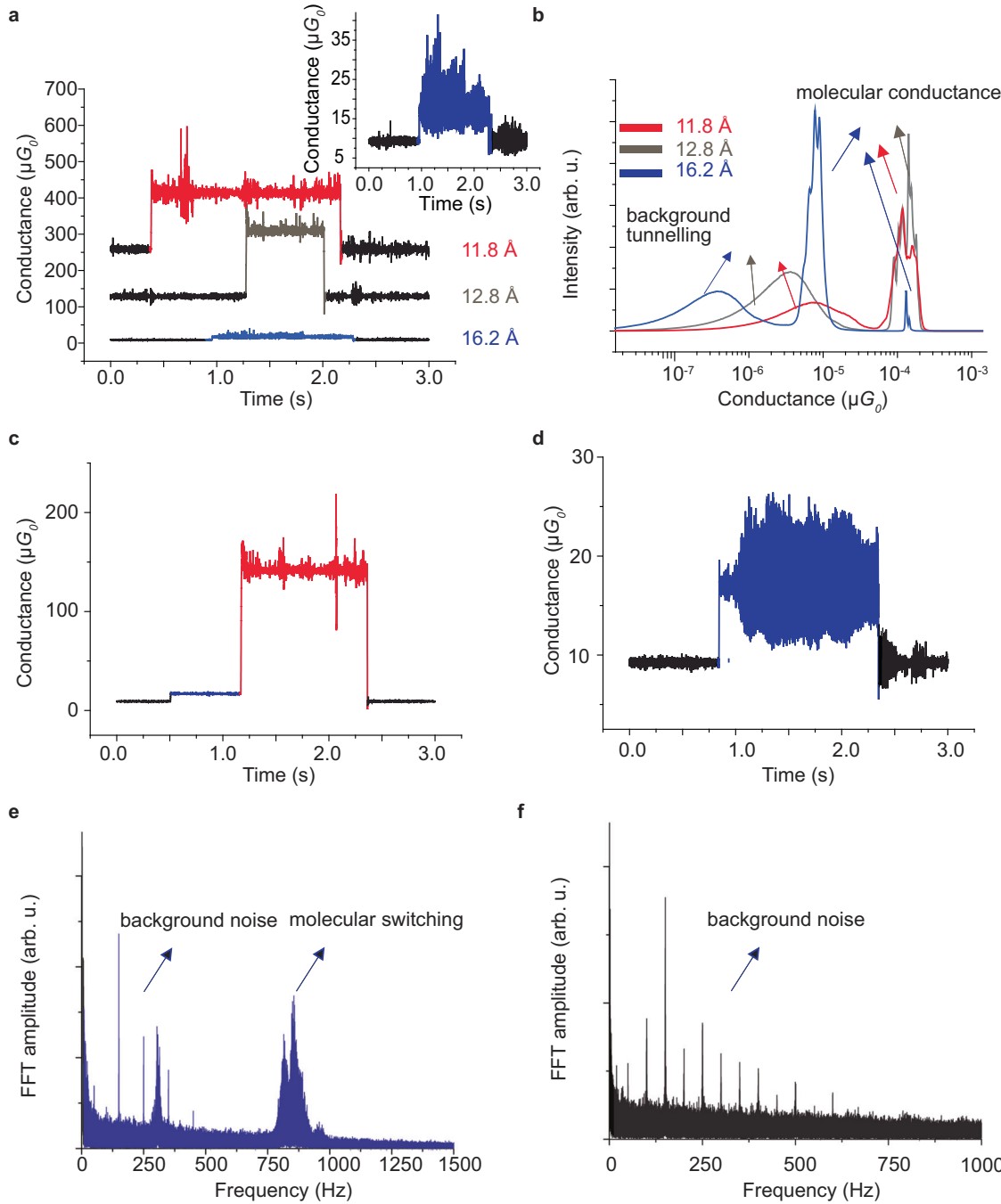

**Fig. 4 | STM single-molecule blinking (current–time) experiments.** *Blinking current–time traces* (**a**) representative blinks at specific separation between the tip and the surface at 11.8 Å (red trace), 12.8 Å (brown trace) and 16.2 Å (blue trace). The surface bias was +100 mV. **b** Conductance histograms built from 100 blinks for each distance; arrows at low conductance indicate the background tunnelling between the two gold electrodes in the absence of a molecule, whereas arrows at high conductance indicate through-molecule conductance. **c** An example of a blink, which represent 10% of the blinks observed at 16.2 Å, that switches from ≈10 $\mu G_0$ (blue trace) to 100 $\mu G_0$ (red trace). The molecular conductance is the difference between the blinking conductance and the background tunnelling conductance. **d** Typical of 80% of blinks at 16.2 Å, showing the current fluctuating by 2–5 fold at 750–1000 Hz (blue trace), unlike the blinks observed at 11.8 and 12.8 Å. **e** FFT analysis of the blinks observed at 16.2 Å assigned to bullvalene switching (blue histogram). **f** FFT analysis of the background tunnelling current in the absence of a connecting molecule.

the long isomer **E**, the intermediate-length isomer **K**, and the short isomer **L** (Supplementary Table 1).

## Conductance modelling of ordered interfaces using NEGF calculations based on DFT KMC simulations

Historically, the modelling of STMBJs has focused on the use of ordered interfaces made by assuming that the gold surfaces formed post collision relax on the available (μs) timescale to form stable regular tips to which molecules bind. As just one tip profile is commonly used, these models typically do not account for the width (large $\eta$) of observed conductance histograms, but they do account for observed variations in peak maxima with changes in molecule and contact composition and are hence anticipated to account for the effects of bullvalene isomerisation on conductance.

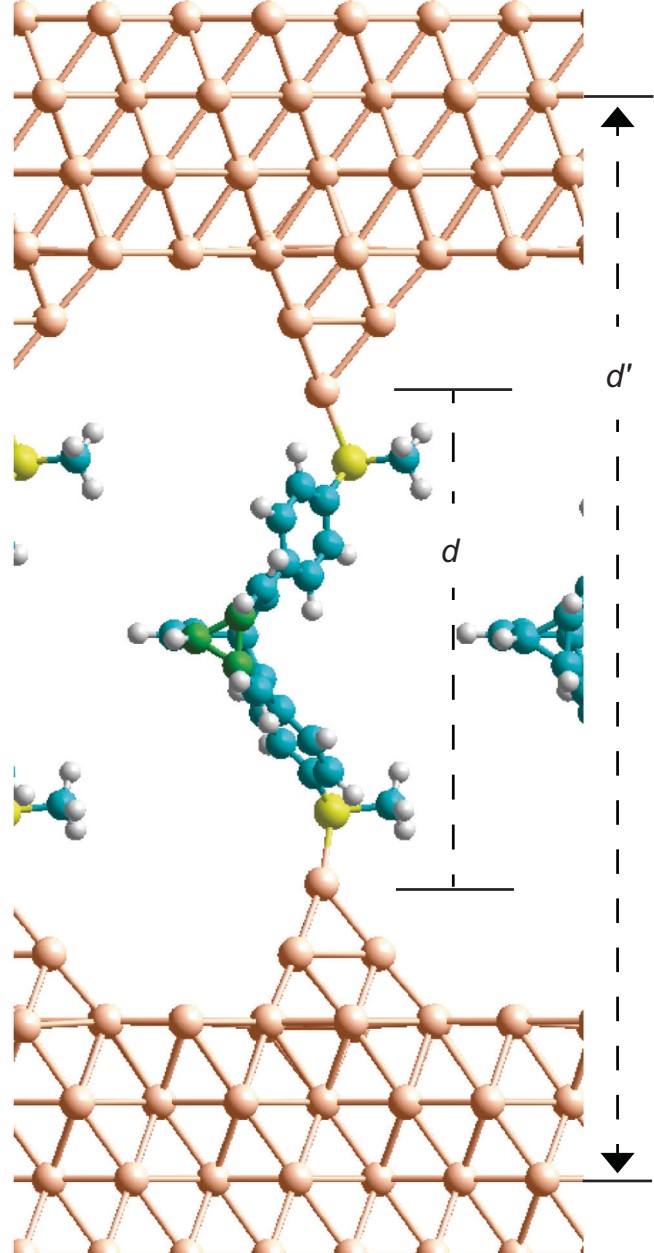

**Fig. 5 | Tipped gold – diaryl bullvalene – gold junctions used in initial STMBJ simulations.** Tipped gold – diaryl bullvalene – gold junctions comprising of 4-layer Au(111) (4 × 4) surfaces plus 4-atom atomic models to which bullvalenes bind. The structure shown is for the $A_{mm}$ conformer at $d = 15.2$ Å; green- bullvalene 3-membered ring, cyan- other carbon, white- hydrogen, yellow- sulfur, gold- gold. The parameter $d$ is frozen in initial cluster-model simulations, whilst $d'$ is frozen in these periodic-image simulations, along with the outer two gold layers of each electrode model. Source data are provided as a Source Data file.

Simulations of diarylbullvalenes made using such a periodic-slab model of the gold contacts, with 4-atom gold tips binding the molecules, are exampled in Fig. 5. Cluster-model calculations are used in preliminary optimisations, with the apex-apex vertical distance $d$ frozen, and then these structures are embedded into a 2D-periodic 4-layer Au-slab model with the two outer layers frozen at distance $d'$. As $d$ is more descriptive of junctions, its (then approximate) value is reported. In the figure, the periodic images of the tips and bound molecules can be seen to be well separated, with the system examplified being the **mm** conformer of the **A** bullvalene constitutional isomer, $A_{mm}$, at $d = 15.2$ Å.

The 3-membered ring of the bullvalene is highlighted in green at the image centre.

Critical aspects of the conductance-trace simulations are highlighted in Fig. 6, with full details given in Supplementary Information. First, in Fig. 6a potential-energy surfaces as a function of $d$ are shown for the constitutional isomers **A** – **L** (except only single points are shown for the high-energy isomers **F, I, J,** and **L**). For the low-energy isomers **A** – **C**, surfaces were constructed for all four conformers **mm, mp, pm,** and **pp**, with the energy of the lowest-energy conformer predicted at each distance only presented in the figure; see Supplementary Table 2 for all data. Conformation energy differences can exceed 0.2 eV at small $d$, but become small above ≈15.6 Å as the molecules are then under tensile strain. For the remainder, potential-energy surfaces were constructed for the most relevant conformers only. For $d < 15.6$ Å, **A** is predicted to be the dominant isomer, forming tight π-stacked ($A_{mp}$) and T-shaped ($A_{mm}$) inter-aryl interactions at short distances down to $d ≈ 9$ Å. Above 15.6 Å, the longer **B** and **C** isomers dominate, joined by **E** above 19 Å.

In Fig. 6b and Supplementary Table 3 are shown calculated transition-state energies for Cope rearrangements involving the most significant constitutional isomers, for their **mm** conformers. Most important are the $A_{mm}$-$D_{mm}$, $B_{mm}$-$D_{mm}$, $B_{mm}$-$C_{mm}$, and $B_{mm}$-$E_{mm}$ transition states as these depict interconversion of the lowest-energy constitutional isomers. From this data, combined with the isomer energetics from Fig. 6a and Supplementary Table 2, is calculated the relative reaction rate compared to that predicted by analogous calculations for bullvalene isomerisation in solution. This is represented in Fig. 6c as calculated speedup factors for the Cope rearrangements, which are then applied to the observed[40] Cope-rearrangement rate in solution of 1.1 kHz per reaction. Note that the rates for forward and reverse reactions will usually differ, and the speedup factors shown are for the slowest of these reactions, with the other process varying from similar rates to rates on the ps timescale. The slowest rates are therefore indicative of the timescale needed to complete oscillating reactions, which from Fig. 6c are predicted to be sped-up or slowed-down by factors of up to 1000. These predictions arise as the gold contacts affect each isomer and transition-states differently, owing to the aryl attachments applying forces at different orientations to the bonding network. Consequentially, transition states can be stabilised or destabilised significantly with respect to the reactants; the speeding up of reactions by straining transition states is a key principle of both enzyme catalysis and materials property modulation, and has been noted previously in junction simulations[33].

Using the calculated reaction rates, chemical kinetics equations are then solved, given an initial $A_{mm}$ conformation at short tip separations, as a function of tip retraction at the rate used in the experiments of 0.5 Å ms⁻¹. Results are shown in Fig. 6d (for isomers predicted later to contribute significantly to conductance traces) and Supplementary Fig. 15 (for other isomers). In addition to the isomerisation reactions, junction breakage is included, using a calculated dissociation energy of 1.3 eV. Note that, at dissociation, the force on the junction is predicted to be ≈0.5 eV Å⁻¹, in good agreement with observed forces for $SCH_3$ – bound gold junctions[56] of 0.44 eV Å⁻¹ (0.7 nN). In summary, the dominant constitutional isomers are predicted to change from **A** at short apex-apex distances to **B** and **C** at intermediate distances to **E** just before dissociation.

Next, Fig. 6e and Supplementary Table 4 shows conductances calculated by NEGF at the DFT-optimised isomer junction geometries. Significant differences in conductance based on bullvalene constitutional isomerism are predicted, but the predicted effect of conformational isomerism is much larger, with up to thousand-fold changes noted. For some isomers, conductances are predicted to also depend significantly upon the apex-apex distance $d$. Transmissions calculated by NEGF are shown in Supplementary Fig. 14. They all show a similar pattern and are suggestive that single conductance pathways,

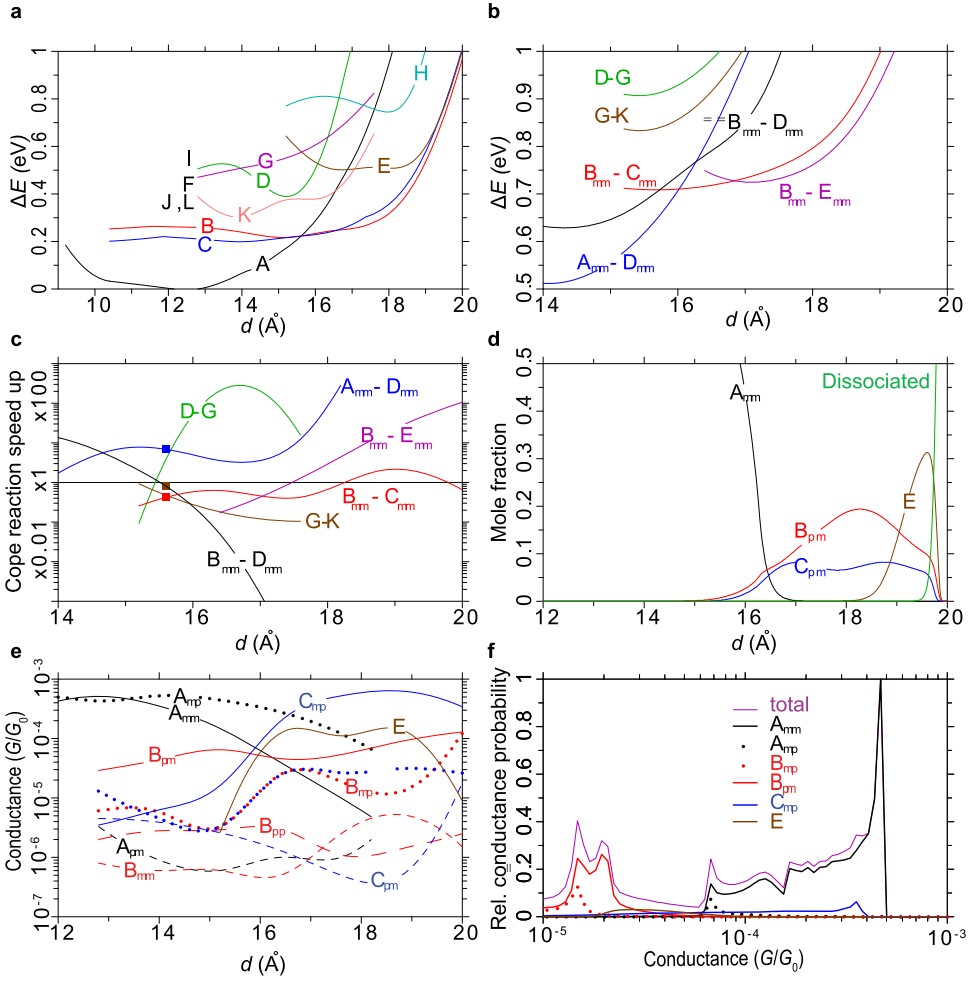

**Fig. 6 | STMBJ kinetic Monte Carlo simulations of tipped gold – diaryl bullvalene – gold junctions.** STMBJ kinetic Monte Carlo simulations of tipped gold – diaryl bullvalene – gold junctions retracted at 0.5 Å ms$^{-1}$ and sampled at 30 kHz for bullvalene constitutional isomers **A – L** at various conformations **m** or **p** of the bullvalene-aryl bonds. **a** Lowest-conformer DFT potential-energy surfaces, or else just single-point energies for **F, I, J,** and **L. b** Various analogous DFT transition-state potential-energy surfaces. **c** Speedup factor for various Cope-rearrangements in situ for the slowest interconversion reaction rate, compared to that of a single reaction in solution[40] (observed[40] at 1.1 kHz); the squares pertain to the **A – D – B** interconversion at the equipotential separation of $d = 15.6$ Å. **d** Mole fraction of highly conductive isomers (results for others are shown in Supplementary Fig. 6). **e** NEGF junction conductances for the most significant isomers. **f** Conductance histogram made from 3000 simulated traces, and its major isomeric contributors. Colours for isomers: **A** black, **B** red, **C** blue, **E** brown. Colours for transition states: **A–D** blue, **B–C** red, **B–D** black, **B–E** magenta, **D–G** green, **G–K** brown. Source data are provided as a Source Data file.

rather than interfering channels, dominate bullvalene constitutional effects.

Finally, conductance histograms are simulated and shown in Fig. 6f. These are obtained by directly mimicking the experimental procedure in which conductance is sampled at 0.033 ms intervals as the tip is retracted. KMC calculations are performed that simulate 3000 individual conductance traces and then average their properties. Each trace starts with the **A**$_{mm}$ conformer at $d = 9.2$ Å, using a time step of 11 ns. At each time step, the reaction rates are determined (Fig. 6c) and a new isomer assigned using a random number generator weighted by the relative rates. Using Fig. 6e, the conductance is then determined and averaged over the 3000 time steps comprising each 0.033 ms sampling interval. In this way, single-molecule conductance is attributed to mixtures of isomers that are in equilibrium on the timescale of the sampling period, with equilibration processes requiring longer timescales being excluded. As calculated isomeric conductances vary by a factor of 1000 (Fig. 6e), in general, small amounts of highly conducting isomers present in the equilibrium mixtures dominate the calculated conductance. Note that the mole fractions of each isomer at each $d$ may be calculated by averaging over

the 3000 trajectories (Supplementary Fig. 6), giving results that agree with the direct kinetics simulations already reported in Fig. 6d.

The simulated conductance histogram (Fig. 6f) displays many features reminiscent of the experimentally observed one (Fig. 3c). There is an initial high-conductance spike typical of most applications for STMBJ calculations made using a 4-atom-tip model, owing to neglect of junction heterogeneity. This dominant peak is attributed to junctions adopting mostly either the **A**$_{mm}$ isomer (represented including contributions from its symmetrically equivalent **A**$_{pp}$ form). Also, the simulated histogram shows two sharp low-conductance peaks that are atypical of molecular junctions yet identified experimentally in Fig. 3c. They are attributed to bullvalene isomerisation and associated ordering of the interface geometry under tension and owe to small amounts of the highly conductive isomer **B**$_{pm}$ contained in the equilibrium isomeric mixtures that exist during the 0.033 ms data-sampling periods.

Nevertheless, two predicted qualitative features differ from those observed: (i) no calculated traces indicate dissociation before isomerisation whereas >80% are observed to, and (ii) individual conductance traces show frequent reversible isomerisation, whereas this

is rare in the observed traces. Supplementary Note 2 shows that these effects are predicted to be sensitive to the experimental parameters (retraction rate and sampling time) used. Three simple modifications to the DFT-calculated isomer and isomerisation energies are considered therein that all alleviate effect (ii) by slowing reactions enough to block reverse processes. By increasing the activation energy for **A** to **D** conversion by 0.10–0.13 eV, the third scheme predicts that 70%–90% of conductance traces should dissociate before isomerisation. Nevertheless, the calculations predict that the rate of the **A** to **D** conversion should be increased in junctions under tension (Fig. 5d) rather than significantly slowed, making this interpretation unlikely.

Observed in blinking experiments at a junction separation of $d = 16.2$ Å, Fig. 4c shows a jump from a conductance state at 8 $\mu G_0$ to one at 150 $\mu G_0$, whereas Fig. 4d shows an oscillating reaction at ≈850 Hz between two isomers at ≈12 $\mu G_0$ to one at 24 $\mu G_0$. These results may be interpreted by adapting the computed reaction rates for the STMBJ experiment at $d = 15.6$ Å highlighted by the squares in Fig. 6c. The peak at 150 $\mu G_0$ is rationalised as arising from isomer **A$_{mm}$** under tension, which can react to form equilibrium mixtures of **B$_{pm}$** and other isomers at much lower conductance. The 850 Hz reaction is attributed to this equilibrium process, predicted by the calculations to occur at 450 Hz (from Fig. 6c and the observed solution isomerisation rate).

## Conductance dynamics modelling after bullvalene-driven interface restructuring

As the calculations using a 4-atom tip model do not support junctions dissociating before isomerisation, alternate binding possibilities are considered. The 4-atom-tip model assumes that the tip atoms coalesce on the μs timescale after junction fracture but remain kinetically trapped in that form on the time scale of the STMBJ experiment. For an isolated Au(111) surface with a 4-atom tip, DFT predicts an activation energy of 0.34 eV for the transfer of the apex atom back one row towards the surface, which, based on the transition-state theory, would be expected to give reaction rates on the 100 ns timescale. 4-atom tips would therefore rearrange before they could be detected by STMBJ, except if the bridging molecule binds faster and inhibits the rearrangement. MD simulations of alkanedithiol junctions[47] indicate that molecules can bind to gold even before the gold-gold contacts break, with the malleable molecules like alkanes becoming largely extended before breakage so that junctions do not form initially with low-conductance gauche conformers[47,48]. Hence molecular binding could indeed inhibit tip restructuring, leading to the wide-ranging success[15,16,50,51] of 4-atom-tip models for the interpretation of STMBJ data.

The binding of diaryl bullvalenes is expected to be significantly different from both that for malleable alkanes and for typical rigid molecular linkers, owing to their ability to bind over a very wide length range. This happens not only through constitutional isomerism, but also through the ability of **A** conformers to compress tightly by utilising strong intramolecular aryl-aryl van der Waals interactions (Fig. 5b). In Fig. 7, optimised structures are presented for simulations made with 3-layer Au(111) (5 × 5) surface models with bound 10-atom tips. This model allows for full compression of the bullvalenes under compressive stress, for extension under tensile stress, slippage of the bullvalenes along the tip sides, and atomic reconstruction of the tips.

In Fig. 7a, the **A$_{mp}$** bullvalene isomer is bound to the sides of two tips at $d = 9.2$ Å, the closest apex-apex distance used in the STMBJ simulations shown in Fig. 6. The molecule binds strongly, with a total interaction energy of −2.40 eV (the dissociation energy for a single optimised Au–S bond was previously calculated to be 1.3 eV). Nevertheless, relaxation over a small barrier leads to the π-stacked dimer shown in Fig. 7b in which the bullvalene is attracted into the cavity between the tips. Such a reaction would be driven by the enhanced reactivity of the apex atoms, combined with the strong intramolecular *aryl-aryl* interaction. This structure was then pushed over a transition state at a relative energy of 0.47 eV (Fig. 7c) to facilitate a cascade of low-barrier chemical reactions leading to the intermediate shown in Fig. 7d in which one tip has been reconstructed, and then on to the final structure shown in Fig. 7e in which both tips are reconstructed. This structure is 2 eV more stable than the original fully tipped structure (Fig. 7b), indicating that a strong driving force exists for the reconstruction of apex-shaped tip models. Restructuring of tips has also been seen in simulations of bound dithioalkanes to gold[47].

As mentioned before, similar calculations performed for bare tips predict a barrier to tip reconstruction of only 0.34 eV. Such low barriers would facilitate tip reconstruction before any conductance measurements could be made using current STMBJ apparatus, suggesting that 4-atom tip models are generally unrealistic. This effect has been seen[47] in ps-timescale simulations of gold tips bridged by alkanedithiols. Binding of **A$_{mp}$** is predicted to increase the barrier and hence increase the reconstruction time to 0.01 ms, but this time remains less than the experimental data sampling interval of 0.033 ms.

The structure of **A** bound to the reconstructed tip in Fig. 7e was then taken, the inter-electrode distance shortened by one Au atomic spacing, and many conformers of **A − E** optimised, see Supplementary Table 5. Equal lowest-energy structures were found to be conformers of **C** and **E**. Given the large amount of energy released as the tip reconstructs, it is reasoned that **A** isomers cannot be found in

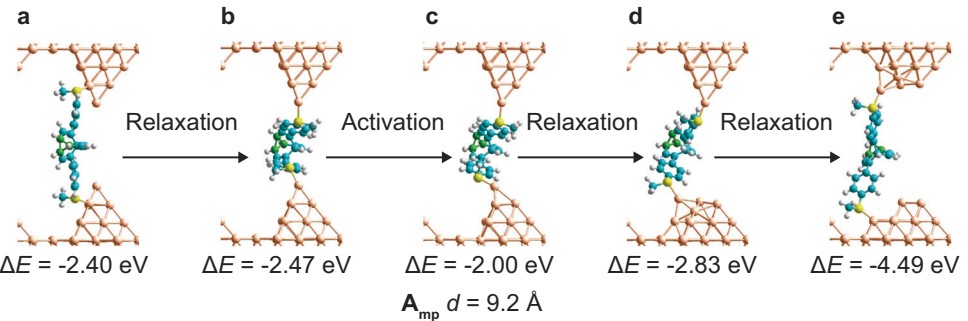

| a | b | c | d | e |
|---|---|---|---|---|
| $\Delta E$ = -2.40 eV | $\Delta E$ = -2.47 eV | $\Delta E$ = -2.00 eV | $\Delta E$ = -2.83 eV | $\Delta E$ = -4.49 eV |

**A$_{mp}$** $d = 9.2$ Å

**Fig. 7 | Bullvalene-driven tip reconstruction.** At $d = 9.2$ Å, an **A$_{mp}$** bullvalene conformer is bound to two gold tips with interaction energies $\Delta E$ with respect to separated tips and molecule: **a** initial extended structure bound to tip sides; **b** after relaxation to make a π-stacked structure located within the tip gap; **c** is activated to make a transition-state for tip reconstructing; **d** a cascade of processes leading to flattening of the lower tip; **e** a cascade of processes leading to flattening of the top tip. The atomic model used is a 3-layer Au(111) (5 × 5) surface with 10-atom tips; green- bullvalene 3-membered ring, cyan- other carbon, white- hydrogen, yellow-sulfur, gold- gold. Source data are provided as a Source Data file.

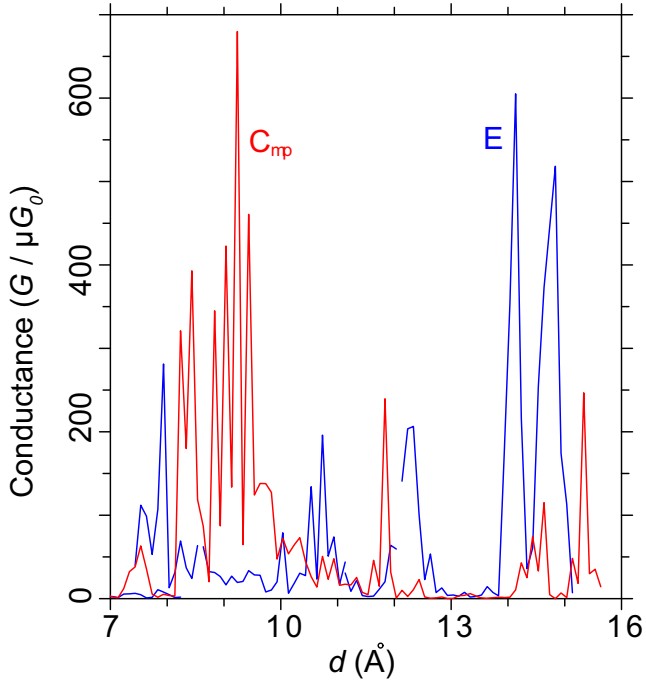

**Fig. 8 | Simulated conductance traces for collapsed tips.** The collapsed structures for $C_{mp}$ (red trace) and **E** (blue trace) optimised at $d = 6.8$ Å are retracted using MD at a rate of 1 Å ps$^{-1}$ and the conductance calculated (exposing sub-ps-timescale dependence of conductance on atomic vibration). The strong conductance predicted at $d = 8$–10 Å could account for the observed average 100 $\mu G_0$ observed in Fig. 3c, whilst the conductance at greater extension could account for the observed 9 and 13 $\mu G_0$ peaks. Source data are provided as a Source Data file.

reconstructed junctions. Hence conductance changes resulting from constitutional isomerism, as revealed in Fig. 6 for regular tipped junctions, are not expected. Figure 8 shows NEGF conductances calculated by taking the optimised **C** and **E** structures and retracting the tip[19,47] using MD simulations at 1 Å ps$^{-1}$. Such rapid retractions fail to average over atomic vibrations, junction rearrangements, etc., and hence show large conductance variations, but averaged properties are likely to be reasonably represented[19]. Some structures sketching the time evolution of **C** are shown in Supplementary Fig. 16, with further details of both calculations provided in Supplementary Figs 17 and 18. At times, these rapidly retracted structures get close to junction breakage, with conformational isomerism occurring spontaneously. Qualitatively, the results for **C** present conductances typical of those observed in the anomalous conductance traces in which conductance is observed to start high and then switch to one or two low-conductance forms.

In summary, single diarylbullvalene molecules are observed to undergo constitutional and configurational isomerisation inside an STMBJ, controlled by adjusting the distance between the attached gold contacts. Hence a NEMS piezoresistance device is constructed that is stable on length scales of 3–9 Å that can display factor of 10 changes in conductance. It is anticipated that much larger conductance changes, perhaps exceeding those seen from just pure conformational isomerism[26,31,32,34,35], can be achieved in the future by introducing synthetic control over conformation. In addition, these chemical reactions are shown to facilitate unprecedented in-situ single molecule switching at ≈850 Hz that is stable on the s timescale. This switching frequency is comparable to that observed in commercial semiconductor transistors (e.g., MOSFETs). Shape-shifting bullvalene molecules therefore present many prospects for the development of measurement and control tools for NEMS systems functioning on the Å atomic-length scale.

A significant property of the bullvalenes is that they can isomerise on the ms timescale to take on vastly different shapes, but this leads to a great many possible atomic-scale data interpretations. The developed DFT/NEGF KMC simulation methodology, and the adapted DFT/NEGF MD methodology, when applied to the prediction of electrode surface structure, molecular binding, molecular isomerisation, and junction conductance, are found to be able to qualitatively interpret all of the major observed features. This provides atomic-level structural and chemical models that can be used to understand function, predict improvements and design new experiments. A basis is therefore provided for the rational improvement of this NEMS technology.

As an example of what this technology has to offer for NEMS devices, two characteristically different trace types were found in the STMBJ experiment. This revealed previously inaccessibly information concerning how NEMS interfaces form and vary during experiments. It is reasoned that one type of trace, formed when the bullvalenes stabilise tipped electrode surfaces, is typical of previously assumed STMBJ structures and facilitates isomerisation-driven piezoresestivity. In contrast, the other type of trace is reasoned to be associated with tip flattening induced by bullvalene constitutional isomerism. Tipped surfaces are thermodynamically unstable yet they are believed to dominate the conventional STMBJ structure, and hence the NEMS experiments reveal features of general relevance to molecular electronics applications. In general, the tipped-interface model for STMBJs that have dominated the field for the last two decades needs to be re-evaluated.

## Methods
### Synthesis
NMR spectra were recorded using an Agilent 5600 MHz DD2 console with an Oxford 600 MHz magnet and Agilent cryoprobe, or Bruker UltraShield Avance III 600 MHz NMR Spectrometer. Residual solvent peaks were used as an internal reference for $^1$H NMR spectra [CDCl$_3$ δ 7.26 ppm] and for $^{13}$C NMR spectra [CDCl$_3$ δ 77.16 ppm]. Coupling constants ($J$) are quoted to the nearest 0.1 Hz. The following abbreviations, or combinations thereof, were used to describe $^1$H NMR multiplicities: s = singlet, d = doublet, t = triplet, q = quartet, m = multiplet, ap. = apparent, br. = broad. IR spectra were recorded neat on a Perkin Elmer spectrum 100 FT-IR Spectrometer with an ATR. ESI-HRMS measurements (electrospray ionisation - ESI) were recorded on an Agilent 6230 time-of-flight LC/MS system. The bis-parathioanisole bullvalene was dissolved in acetonitrile with silver nitrate 0.1% w/v, whereby the [M + Ag]$^+$ value is quoted to Ag$^{107}$. Melting points were recorded on a DigiMelt melting point apparatus, model MPA161 and were not corrected. Analytical thin-layer chromatography was conducted with commercial aluminium sheets coated with 0.25 mm silica gel (E. Merck, silica gel 60 F254). Compounds were either visualized under UV-light at 254 nm, or by dipping the plates in aqueous potassium permanganate or cerium ammonium molybdenite solution followed by heating. *Rf* values were recorded to the nearest 0.05 cm.

Synthesis of bis(4-thioanisole)bullvalene:

To a solution of bis-Bpin-bullvalene (20 mg, 0.052 mmol, 1 eq.), Pd$_2$(dba)$_3$ (10 mg, 20 mol %) and [HP(t-Bu)$_3$]BF$_4$ (12 mg, 80 mol %) in degassed THF (2.5 mL) was added a degassed aqueous 10% NaOH (0.1 mL) solution and stirred for 2 min. 4-bromothioanisole (23 mg, 0.052 mmol, 2.2 eq.) was then added and the solution stirred for 3 h at 65 °C. The reaction mixture was taken up in water and extracted with DCM (x3). The organics were combined, washed with brine, dried over MgSO$_4$ and concentrated under rotary evaporation. The crude product was purified by silica gel flash column chromatography (0–30% EtOAc/Hex) and then recrystallised from DCM/Hex (1:3) to give the product as a yellow solid (16 mg, 82%).

R$_f$: 0.50 (EtOAc/Hex 30:70); m.p. 140–141 °C; $^1$H NMR (600 MHz, CDCl$_3$, −40 °C) δ 7.33 (d, J$_{HH}$ = 8.2 Hz, 1.72H), 7.29 (d, J$_{HH}$ = 8.2 Hz,

1.34H), 7.21–7.17 (m, 3.88H), 7.15 (d, $J_{HH}$ = 13.1 Hz, 6.08H), 7.13 (s, 1.18H), 6.22–6.19 (m, 2.08H), 6.17 (d, $J_{HH}$ = 8.3 Hz, 0.64H), 6.11–6.01 (m, 3.79H), 5.95 (d, $J_{HH}$ = 11.4 Hz, 0.67H), 3.53 (d, $J_{HH}$ = 8.3 Hz, 1.00H), 3.19 (t, $J_{HH}$ = 9.0 Hz, 0.69H), 2.84 (d, $J_{HH}$ = 8.8 Hz, 0.23H), 2.79 (t, $J_{HH}$ = 8.6 Hz, 0.65H), 2.62–2.35 (m, 15.65H) ppm; $^{13}$C NMR (150 MHz, CDCl$_3$, −40 °C): δ = 141.6 (C), 139.8(C), 139.6 (C), 139.3(C), 139.2 (C), 138.3 (C), 136.5 (C), 136.3, 136.3 (C), 128.1 (CH), 127.4 (CH), 126.8 (CH), 126.8 (CH), 126.7 (CH), 126.3 (CH), 126.2 (CH), 125.8 (CH), 125.8 (CH), 125.7 (CH), 124.3 (CH), 123.6 (CH), 122.6 (CH), 39.9 (CH, isomer A), 35.1 (CH, isomer B), 29.9 (CH, isomer C), 23.4 (CH), 21.8 (CH), 21.7 (CH), 20.91 (CH), 20.80 (CH), 15.67 (CH$_3$, isomer B), 15.65 (CH$_3$, isomer C), 15.62 (CH$_3$, isomer B), 15.48 (CH$_3$, isomer A). IR (ATR): $\bar{v}$ = 3029, 2922, 1491, 1095, 909, 863, 819, 785, 769, 742 cm$^{-1}$; HRMS (ESI): $m/z$ calcd. for C$_{24}$H$_{22}$S$_2$Ag$^+$: 481.0208 $[M + Ag]^+$, found: 481.0195. NMR spectra are found in the Supplementary Information.

Synthesis of para - (4,4'-bi(methylthio)terphenyl):

This compound was synthesised according to a slightly modified procedure taken from ref. To a solution of 4-(methylthio)phenylboronic acid (0.24 g, 1.4 mmol), 1,4-dibromobenzene (0.17 g, 0.7 mmol) and Na$_2$CO$_3$ (0.38 g, 3.8 mmol) in degassed toluene/water (9.8/1.2 mL) was added Pd(PPh$_3$)$_4$ (0.08 g, 0.07 mmol). The solution was stirred and heated at 100 °C for 16 h. The reaction solution was cooled to room temperature, diluted in MeOH, and filtered. The remaining solids were washed with water, then 1 M HCl, followed by MeOH (x4) to afford (4,4'-bi(methylthio)terphenyl) (0.08 g, 35%) as a cream white solid

R$_f$: 0.50 (50% DCM/Hex); $^1$H NMR (500 MHz, CDCl$_3$) δ 7.64 (s, 1H), 7.57 (d, $J_{HH}$ = 8.4 Hz, 1H), 7.35 (d, $J_{HH}$ = 8.3 Hz, 1H), 2.54 (s, 1H) ppm; $^{13}$C NMR (150 MHz, CDCl$_3$) δ 139.5, 137.9, 137.6, 127.5, 127.4, 127.2, 16.1 ppm; MP > 260 °C; HRMS (ESI): m/z calcd. for C$_{20}$H$_{18}$S$_2^+$: 322.08499 [M]+, found: 322.0844.. NMR spectra are found in the Supplementary Information.

## STMBJ measurements

The STM measurements were carried out at room temperature in 1,2,4 − trichlorobenzene solvent using 50 μM solution of the diaryl bullvalene with a PicoSPM I microscope head controlled by Picoscan 2500 electronics from Agilent. The STMBJ data were collected using NI-DAQmx/BNC-2110 national instruments (LabVIEW data collection system) and analysed with a home-made code based on the LabVIEW software. Around 3000–6000 current versus distance curves were collected and accumulated into a conductance histogram. Curves displaying clear plateaus that evidence the formation of a single-molecule bridge were automatically selected and used to build the conductance histograms. An automatic selection driven by a home-made Labview code was used to select the individual traces. The histograms were made by applying the same automated selection criteria to each set of recorded decay curves. The percentage decay curves that showed clear plateaus displaying the occurrence of a single-molecule bridge were typically 40–50% and were all selected to build the histograms. This selection process made peaks in the conductance histograms more prominent above the tunnelling background. Untreated histograms, i.e., without any selection, are presented in the Supplementary Information.

## STMBJ blinking approach measurements

These measurements were made on the same equipment used for the STMBJ experiments. In the blinking approach, the tunnelling current is first stabilised for at least 1 h until a tunnelling current variation of <10% is obtained. Current transients are then captured when a molecule connects between the STM tip and the surface in the presence of a 5 μM solution of the diaryl bullvalene in 1,2,4 − trichlorobenzene. More details of the procedure and its interpretation appear in Supplementary Note 1.

## DFT calculations

For the calculations using regular 4-atom Au tips, initially optimised structures and transition states were determined using a cluster model. In this, the 2D gold surfaces were ignored, with four frozen atoms in a triplet state used simply to represent the tips. The apex-apex distance was frozen at distance $d$, evaluated on a grid from $d$ = 9.2 to 20 Å, spaced at 0.2 Å. The cluster calculations were performed by Gaussian16[57] using the PBE density functional[58] and Goerigk and Grimme's "D3(BJ)" empirical dispersion correction[59]; the 6–31 G* basis set[60] was used for S, C, and H, with LANL2DZ[61] used for Au. Transition states were found by manual grid searching and interpolation utilising the two changing C-C bond lengths. At least every sixth resulting structure was then transferred in between two (4 × 4) 2D periodic models of the Au(111) surface made using an Au-Au bond length of 2.897 Å. large vacuum region was used to separate images perpendicular to the slabs, using a lattice constant of 55 Å. Geometry optimisations were then performed using VASP 5.4[62], using PAW pseudopotentials[63], high precision, and a basis-set energy cutoff of 503 eV. The PBE density functional was used, with D3(BJ) corrections replicated within the 2D plane only. The outside two layers of each slab were frozen during this process, freezing the distance $d'$ (Fig. 5). This distance relates to that frozen in the cluster calculations through $d' = d + 18.066$ Å, with the apex-apex distance $d$ allowed to vary slightly throughout the ensuing optimisation. Other VASP calculations pertaining to gas-phase molecules and to tipped gold structures without bridging molecules were similarly performed. Resulting optimised coordinates, calculated energies, and key computational parameters are provided in full in Supplementary Information.

Calculations for the tip restructuring (Figs. 7 and 8) were performed using a similar 2D modelling approach, except that a (5 × 5) 2D periodic model was used containing three complete gold layers, with 10-atom tips. The MD simulations were performed using "ALGO = VERYFAST" and "PREC = LOW" at 300 K using a time step of 0.1 fs, with the tip retracted at a rate of 1 Å ps$^{-1}$. As indicated in Supplementary Figs 3 and 4, sometimes these trajectories headed toward junction dissociation, requiring some manual enactment of isomerisation and/or interfacial rearrangement reactions expected to occur on ns timescales much larger than these simulations allow for. The Supplementary Information lists key optimised and transition-state structures, as well as one structure from each structural domain of the MD simulations.

## NEGF calculations

The NEGF calculations were performed using Nanodcal[44]. The molecular electronic junction was divided into three regions, bottom electrode, contact region, and top electrode. The contact region includes parts of the physical electrodes (4 (111) layers of gold), additional electrode atoms to model tip structure, and the bridging bullvalene molecule. All NEGF calculations of molecular junction conductances were performed self-consistently. Used was the PBE density functional, double-zeta with polarisation basis set, **k**-space grids of 3 × 3 × 1, and 50 a. u. energy cut-off for the real-space grid.

## KMC simulations

These were performed by the programme "kinetics_t.for" using input data (from Supplementary Tables 3–5) listed in "surface_energies.csv", with both files available on Zenodo[64]. They solve for the time evolution of the isomeric composition, given the calculated isomeric energies, Cope-reaction transition-state energies, and isomeric conductances, with the tip being constantly retracted throughout the simulations. A small correction was applied to all of the calculated transition-state energies so that the calculated value for bullvalene matched the observed Cope-reaction rate in solution (3.2 kHz net for the 3 possible Cope rearrangements). This approach ignores the change in

solvent from $CCl_2=CCl_2$ ($\epsilon=2.268$) for the kinetics measurements to TCB ($\epsilon=2.24$) for the STMBJ measurements. The temperature was set to 300 K. The mole fractions of the **mm**, **mp**, **pm**, and **pp** conformers of **A** – **E** were calculated, as well as that for single conformers of **G** and **K**. Conformational equilibrium is assumed throughout the simulations, which focus on the Cope rearrangements only. Transition-state energies were calculated for one conformer only, these being used to approximate the energies for the other conformers. The raw calculated data was interpolated at different values of $d$ using cubic spline functions. The time step used was 11 ns, with reactions presenting rate constants on a shorter timescale being represented simply as chemical equilibria. Monte Carlo methods were used to assign a single isomer in a conductance trace during each time step, with 3000 time steps averaged over to simulate single data-point current measurements performed using a sampling time of 0.033 ms. See Supplementary Note 3 for more details.

### Reporting summary

Further information on research design is available in the Nature Portfolio Reporting Summary linked to this article.

## Data availability

The data that support the findings of this study are available from the corresponding authors upon request. STM experimental data is available from N.D. upon request. To ensure proper accessibility, the data will be converted from a binary format to a text format. We will be pleased to perform this conversion along with providing the necessary calibration specifications upon request. Synthetic methods and NMR spectra are available from T.F. NEGF calculation results are available from D.S.K. Source data are provided with this paper.

## Code availability

The single-molecule data were acquired using a custom software (Labview, National Instruments) and are available from A.C.A. and N.D. upon request. The kinetics Monte Carlo simulation code and data are available on Zenodo[64].

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

## Acknowledgements

We thank the Australian Research Council for support through grants (DE160101101 and DP190100735). We also thank the National Computational Infrastructure (NCI) Australia for computational support through grant d63, as well as the Shanghai University ICQMS high-performance computing facility. A.C.A. thanks to the Generalitat de Catalunya the Beatriu de Pinós Programme and to the Spanish MICINN for the MDM-2017-0767.

## Author contributions

N.D. and T.F. conceived the idea and initiated the project. N.D. designed and executed the single-molecule experiments with contributions from T. Li., A.C.A. performed the correlation maps and the FTT blinking analysis. T.F. designed and executed the synthetic work and supervised A.P.B., J.R.R. and D.S.K. designed, executed and supervised the computational work with contributions from L.Y., N.D. and J.R.R wrote the manuscript with significant contributions from D.S.K. and T.F.

## Competing interests

The authors declare no competing interests.
