## [Peer Review file · Nature Communications]

REVIEWER COMMENTS

Reviewer #1 (Remarks to the Author):

In the manuscript entitled “Controlling Single-Molecule Piezoresistivity through Isomerisation of Bullvalenes”, conductance through a single diaryl bullvalene molecule is measured by the STM-BJ technique. The constitutional isomerisation of bullvalene has been demonstrated to construct a single-molecule nano electro-mechanical system (NEMS) piezoresistor with a factor of 10 changes in conductance through stretching experiments, blinking approach experiments and simulations. This work is well done and attractive. These results provide a new insight into the design of NEMS piezoresistors and the understanding of chemical reactions in STM-BJs. However, there are some problems that need to be further improved as follows:

1. In STMBJ stretching experiments, two lower conductance peaks (9 and 13 $\mu\text{G}\Omega$) show bias dependence (line 144). What is the reason for this? Please explain briefly in the manuscript.
2. The trace at 16.2 Å shown in Fig. 4d depicts continuous switching with a switching rate of 850 Hz, while the conductance trace shown in Fig. 4c accounts for 10% of the total number observed and is non-continuous at essentially one level (line 189). What is the approximate switching rate between 8 $\mu\text{G}\Omega$ and 150 $\mu\text{G}\Omega$, or average duration time at 150 $\mu\text{G}\Omega$ in experiments?
3. It is mentioned that “Binding to the gold contacts affects the isomers and their transition states differently, so reaction rates can be sped up, or slowed down, by factors of up to 1000 (line 232)”. What is the possible reason for this? Please explain briefly in the manuscript. Has this conclusion been reported in the literature? It should be better to add the references.
4. Please check the author's affiliation, “3. School of Molecular and Life Sciences, Curtin University, Bentley, WA 6102, Australia.” in the Mian text and “3. School of Molecular and Life Sciences, Curtin Institute of Functional molecules and Interfaces, Curtin University, Bentley, WA 6102, Australia.” in the Supporting Information are different.
5. Some words are in italic in line 277 and 310, please check them.
6. The font of “The bis-parathioanisole bullvalene was dissolved in acetonitrile (Methods, line 378)” should be uniform.
7. The “ μm ” should be change to “ μM ” in “The STM measurements were carried out at room temperature in 1,2,4-trichlorobenzene solvent using 50 μm solution (Methods, line 388)”.
8. The journal name in References is inconsistent, both full journal name and abbreviation exist. Please check and unify.

In summary, this manuscript deserves to be published after minor revisions mentioned above.

Reviewer #2 (Remarks to the Author):

Review of “Controlling Single-Molecule Piezoresistivity through Isomerisation of Bullvalenes” by Reimers et al.

In this paper the authors present experimental and computational studies of bullvalene-based junctions. The authors identify three main conductance features (one of high conductance which is dominant, and two sharp low-conductance features). Through “pull and hold” type of measurements, the authors demonstrate that the two low-conductance states interconvert easily and assign this interconversion to constitutional isomerization. Through computations they explore the complex series of isomers and interconversions that are compatible with this system and offer an interpretation of the effect.

In general, I find that the experiments are careful and, to the extent in which I can determine, well done. The chosen molecule (functionalized bullvalene) is definitively an interesting choice for molecular electronics studies. From my perspective, the most unique aspect of their observations is the isomerization of bullvalene having observable consequences in the conductance at a rate of 850 kHz. The authors argue that the isomerization of bullvalene enables them to have a piezoresistive device that is stable on length scales of 3-9 Å. The computational effort in the paper is quite extensive, and includes quantum chemistry calculations of many possible bullvalene isomers as a function of elongation and a Monte Carlo model of junction evolution.

While the paper is potentially of interest, the paper is not suitable for publication in Nature Communications in its present form.

In particular, the physical picture of what is happening in the experiment is not transparent from the manuscript and the results presented. This makes it difficult to determine with clarity what are the unique contributions by the paper and how it advances the state the art. In fact, after reading the paper several times this reviewer is still confused about the main advances and significant of this paper.

There are two main components that contribute to this confusion:

- i. The writing in the paper is unnecessarily complex, there are several typos and there many open-ended statements that are not justified by discussion or data. Some examples to illustrate this point:
 - a. Line 76, the authors state that the isomerism can modulate quantum interference and that this clearly shown by the data. I did not see any discussion or evidence of the modulation of such quantum interference and its effect on the transport measurements.

b. Line 189, a paragraph break at “In Fig. 4c” would help make the “In contrast” in p. 196 clear. As written, it is very confusing.

c. Line 194 “is mostly not associated with continuous conduction at essentially the one level”. I do not understand the sentence or how it reflects the observations.

d. Line 297 “MD simulations of alkanedithiol junctions perceive”. Perception in its standard use refers to the action of understanding or realizing. I really cannot understand how MD simulations can “perceive”.

e. Line 252, “the observed conductance traces are measured sampling each...” Here the word measured is confusing because it is not helping distinguish the experiments from the computations.

f. The whole paragraph at Line 268-278 is highly convoluted, making it challenging to extract its key scientific content.

g. The paragraph in Line 289 contains many speculative aspects that do not contribute to the clarity in presentations. For example, how are reaction rates on the 100 ns timescale in Line 295 are expected? What is the evidence for molecular binding prevent tip restructuring in Line 300. How do you define “success” for the 4-atom tip model in line 301. These are just some examples, of timescale and energies that are just not justified in the paper.

h. It is unclear what is the evidence for the conclusion in line 364-367.

i. Some of the figures are unnecessarily complex. In particular, this reviewer highly encourages the authors to revisit the Fig. 5 and 6 as they are really hard to read. For example, wouldn't a figure that just focuses on the key species be easier to understand and allow the authors to communicate the chemistry at play? Or, wouldn't sublabel in the different plots help the reader read through the figure? In Fig. 5d it is unclear why the authors decided to use “speed up” instead of actual rates. While the modulation of chemical reactivity by nanoscale environments is of interest, it is not the focus of the discussion where actual rates instead of speed ups is what matters. The difference between f and g is also hard to understand from the figure itself.

j. Fig. 3 and 4 would really benefit from direct assignments of the different features observed that can be easily understood by just looking at the figures. Similarly, I don't think Fig. 1 does a good job at communicating the main physical picture that is emerging.

There are many more examples of this type that, overall, make the paper confusing. In future revisions, the authors should carefully revisit their structure and writing to maximize clarity, fully justify all statements and claims and make their contributions and arguments completely transparent.

ii. While the computations presented represent a solid effort to understand the complexity of the experiment, the picture that emerges from these computations as summarized in Figures 5 and 6 is not clear and, further, it disagrees with experiment. Thus, at the end of the day the computations are not offering a clear atomistic picture that can be used to understand the experiments. It would be advisable for the authors to decide what are reliable computational results that can be used in this paper, and

what part of the computations are still at a level in which the results depend strongly on “ad hoc” undefined parameters and thus not publishable.

Other points:

1. The authors argue that they have a piezoresistor that is stable for several Angstroms. But why is this unique? If I just pull an alkane I will get a piezoresistor and I can choose the length of stability of the piezoresistor simply by changing the length of the alkane. See, for instance:

<https://pubs.acs.org/doi/pdf/10.1021/acs.jpcllett.7b03323>

2. There are other contributions mechanically inducing changes in conductance that may be of interest to the authors.

a. Folding a single-molecule junction: <https://pubs.acs.org/doi/10.1021/acs.nanolett.0c02815>

b. Ring-opening reactions:

<https://pubs.rsc.org/en/content/articlelanding/2019/sc/c8sc04830d>

c. Association reactions:

<https://pubs.acs.org/doi/10.1021/acs.jpcc.1c01901>

d. Control of anchoring:

<https://journals.aps.org/prl/abstract/10.1103/PhysRevLett.121.047701>

e. Control of intermolecular interactions

<https://www.nature.com/articles/nchem.2588>

These papers provide clear examples where isomerism both constitutional and configurational can be used to control transport. In the context of this literature (and related papers) it would be useful for the authors to clearly delineate their advances.

3. The authors use “conductivity” and “piezoresistivity”. However, these quantities refer to intensive quantities that are ill-defined at the single-molecule limit. The correct terms, I would say, are “conductance” and “piezoresistance”.

Reviewer #3 (Remarks to the Author):

I enjoyed reading this manuscript. The choice of molecules for study and the design of the experiments/calculations were very clever. The experiments and calculations appear to have been done thoroughly and competently. The results are new and will be of interest to a wide community of researchers. I recommend that the manuscript be published without significant modifications.

Several small points:

Page 2, paragraph 3: "First, piezoresistivity in demonstrated that is stable..." should be "First, piezoresistivity is demonstrated that is stable".

Page 3, paragraph 3: "...of which B, D, and E each form sets of enantiomeric stereoisomers..." In the caption of Figure 2, the enantiomeric pairs are noted as "(A, A'), (B, B'), and (D, D)'" which is correct?

Page 4, paragraph 2: The authors mention "a non-isomerisable control molecule". It appears from the SI that this is "para - (4,4'-bi(methylthio)terphenyl)". I suggest the authors simply name the control molecule in the text. I don't believe there is a reason not to. This non-isomerisable molecule is mentioned again on p. 5.

Page 4, paragraph 2: "Figure 3b shows that these conductance peaks arise as the tip is retracted to separations too large to support conductance via the mechanism operative for the large peak (red arrow)." The authors' Figures are, of necessity, rather complicated, and each contains a lot of information. As I was reading the text and referring back to the Figures, I started looking for a red arrow in Figure 3b, only to realize (after an embarrassingly long search) the red arrow referred to was the red arrow from Figure 3c. I suggest helping future readers: "Figure 3b shows that these conductance peaks arise as the tip is retracted to separations too large to support conductance via the mechanism operative for the large peak (red arrow in Figure 3c)." I admit that this is a very minor point.

Concerning Figure 3b: There are two faint circles down on the plot, but I could find no reference to them in either the caption or the text. It is easy for the reader to imagine the meaning of these circles, but the authors may want to add a sentence to make the meaning unambiguous.

Page 5, paragraph 2: "In Fig. 4c, an individual conductance trace, representative of 10% of the total number observed, is shown a current trace that starts at the background level, jumps to ca. 8 μG_0 , jumps again to ca. 150 μG_0 ," The text uses units of μG_0 , but the figure uses nA. Wouldn't it be cleaner to use μG_0 throughout?

Page 7, paragraph 3: Reference to Figure 3c should be to Figure 4c.

I congratulate the authors on a splendid piece of science!

Reviewer #1 (Remarks to the Author):

In the manuscript entitled “Controlling Single-Molecule Piezoresistivity through Isomerisation of Bullvalenes”, conductance through a single diaryl bullvalene molecule is measured by the STM-BJ technique. The constitutional isomerisation of bullvalene has been demonstrated to construct a single-molecule nano electro-mechanical system (NEMS) piezoresistor with a factor of 10 changes in conductance through stretching experiments, blinking approach experiments and simulations. This work is well done and attractive. These results provide a new insight into the design of NEMS piezoresistors and the understanding of chemical reactions in STM-BJs. However, there are some problems that need to be further improved as follows:

1. In STMBJ stretching experiments, two lower conductance peaks (9 and 13 μG_0) show bias dependence (line 144). What is the reason for this? Please explain briefly in the manuscript.

Response: we thank the Reviewer for their suggestion. In principle, the electric field strength can restructure junctions, particularly interfaces but for molecules like bullvalenes, internal restructuring could also occur. Conductance is sensitive to such features, and so bias dependence is expected. We did not perceive this as a significant issue, which would have required considerable effort to simulate. To address the Reviewer’s comment, we have added text in the revised manuscript suggesting possible causes.

“The most prominent unexpected peaks are highlighted by blue and black arrows in Fig. 3c; these are reproducible features, but they do show some bias dependence, see e.g. SI Fig. S10. This could arise from the structural changes induced by the electric field.”

2. The trace at 16.2 Å shown in Fig. 4d depicts continuous switching with a switching rate of 850 Hz, while the conductance trace shown in Fig. 4c accounts for 10% of the total number observed and is non-continuous at essentially one level (line 189). What is the approximate switching rate between 8 μG_0 and 150 μG_0 , or average duration time at 150 μG_0 in experiments?

Response: we thank the Reviewer for their suggestion. We have now added in the manuscript page 4 paragraph 2: “the switching rate between 8 μG_0 and 150 μG_0 occurs at a rate of 2 h^{-1} with an average duration of 0.7 s for the blinks at the 150 μG_0 level.”

3. It is mentioned that “Binding to the gold contacts affects the isomers and their transition states differently, so reaction rates can be sped up, or slowed down, by factors of up to 1000

(line 232)”. What is the possible reason for this? Please explain briefly in the manuscript. Has this conclusion been reported in the literature? It should be better to add the references.

Response: we thank the Reviewer for their suggestion. The effect here is one of the principles of enzyme catalysis and also the effect of strain on solid-state reactions, and this is mentioned now in the text. More is also said about the nature of the transition states and the effect that applying forces in different directions via the aryl groups can have. For reactions conducted in molecular junctions, the effect also operates (a reference to this is now added), in parallel to effects induced directly by the electric field (also analogous to enzyme catalysis). All calculations of tip-induced catalysis will implicitly include the effect.

“In Fig. 6b and SI Table S3 are shown calculated transition-state energies for Cope rearrangements involving the most significant constitutional isomers, for their **mm** conformers. Most important are the **A_{mm}-D_{mm}**, **B_{mm}-D_{mm}**, **B_{mm}-C_{mm}**, and **B_{mm}-E_{mm}** transition states as these depict interconversion of the lowest-energy constitutional isomers. From this data, combined with the isomer energetics from Fig. 6a and SI Table S2, is calculated the relative reaction rate compared to that predicted by analogous calculations for bullvalene isomerisation in solution. This is represented in Fig. 6c as calculated speedup factors for the Cope rearrangements, which are then applied to the observed⁴⁰ Cope-rearrangement rate in solution of 1.1 kHz per reaction. Note that the rates for forward and reverse reactions will usually differ, and the speedup factors shown are for the *slowest* of these reactions, with the other process varying from similar rates to rates on the ps timescale. The slowest rates are therefore indicative of the timescale needed to complete oscillating reactions, which from Fig 6c are predicted to be sped-up or slowed-down by factors of up to 1000. These predictions arise as the gold contacts affect each isomer and transition-states differently, owing to the aryl attachments applying forces at different orientations to the bonding network. Consequentially, transition states can be stabilised or destabilised significantly with respect to the reactants; the speeding up of reactions by straining transition states is key principle of both enzyme catalysis and materials property modulation, and has been noted previously in junction simulations³³.”

4. Please check the author's affiliation, “3. School of Molecular and Life Sciences, Curtin University, Bentley, WA 6102, Australia.” in the Mian text and “3. School of Molecular and Life Sciences, Curtin Institute of Functional molecules and Interfaces, Curtin University, Bentley, WA 6102, Australia.” in the Supporting Information are different.

Response: we thank the Reviewer for their careful reading. The affiliations has now been corrected in the revised MS and SI.

5. Some words are in italic in line 277 and 310, please check them.

Response: We thank the Reviewer. The use of italics in those cases is for emphases, stressing unexpected features.

6. The font of “The bis-parathioanisole bullvalene was dissolved in acetonitrile (Methods, line 378)” should be uniform.

Response: this typo has now been corrected.

7. The “ μm ” should be change to “ μM ” in “The STM measurements were carried out at room temperature in 1,2,4-trichlorobenzene solvent using 50 μm solution (Methods, line 388)”.

Response: again, we thank the Reviewer for their careful reading and this typo has now been corrected.

8. The journal name in References is inconsistent, both full journal name and abbreviation exist. Please check and unify.

Response: we appreciate the careful reading from the Reviewer. The references have now been corrected in the revised MS.

In summary, this manuscript deserves to be published after minor revisions mentioned above.

Reviewer #2 (Remarks to the Author):

Review of “Controlling Single-Molecule Piezoresistivity through Isomerisation of Bullvalenes” by Reimers et al.

In this paper the authors present experimental and computational studies of bullvalene-based junctions. The authors identify three main conductance features (one of high conductance which is dominant, and two sharp low-conductance features). Through “pull and hold” type of measurements, the authors demonstrate that the two low-conductance states interconvert easily and assign this interconversion to constitutional isomerization. Through computations they explore the complex series of isomers and interconversions that are compatible with this system and offer an interpretation of the effect.

In general, I find that the experiments are careful and, to the extent in which I can determine, well done. The chosen molecule (functionalized bullvalene) is definitively an interesting choice for molecular electronics studies. From my perspective, the most unique aspect of their observations is the isomerization of bullvalene having observable consequences in the conductance at a rate of 850 kHz. The authors argue that the isomerization of bullvalene enables them to have a piezoresistive device that is stable on length scales of 3-9 Å. The computational effort in the paper is quite extensive, and includes quantum chemistry calculations of many possible bullvalene isomers as a function of elongation and a Monte Carlo model of junction evolution.

Response: we thank the Reviewer for finding our work carefully taken and our system interesting for the field.

While the paper is potentially of interest, the paper is not suitable for publication in Nature Communications in its present form.

In particular, the physical picture of what is happening in the experiment is not transparent from the manuscript and the results presented. This makes it difficult to determine with clarity what are the unique contributions by the paper and how it advances the state the art. In fact, after reading the paper several times this reviewer is still confused about the main advances and significant of this paper.

There are two main components that contribute to this confusion:

i. The writing in the paper is unnecessarily complex, there are several typos and there many open-ended statements that are not justified by discussion or data. Some examples to illustrate this point:

a. Line 76, the authors state that the isomerism can modulate quantum interference and that this clearly shown by the data. I did not see any discussion or evidence of the modulation of such quantum interference and its effect on the transport measurements.

Response: we thank and agree with the Reviewer. This statement is in the introduction and was only meant to describe the features that the calculations can, in principle, include. It has been

rewritten to remove any ambiguity. We plan for a specialised paper on this topic at a later stage. Because of symmetry, all interference effects are off-resonance and so their effects become blurred rather than exposed. In the revised manuscript, we have added to the SI some of most pertinent transmission profiles which are reproduced below. No obvious indication of quantum interference is observed in the transmission spectra. However, not seeing a clear dip in the transmission does not entirely disprove that quantum interference plays a role.

Fig. S14 | NEGF calculated transmissions through Au-diarylbullvalene-Au linkages. The plot shows transmissions as functions of energy away from the Fermi energy computed for different conformers and stretching distances d : a) A_{mm} , b) A_{pm} , c) B_{mm} , d) B_{pm} , e) B_{pm} , f) C_{mm} .

We have also added text in the main manuscript.

“For some isomers, conductances are predicted to also depend significantly upon the apex-apex distance d . Transmissions calculated by NEGF are shown in SI fig. S14. They all show a similar pattern and are suggestive that single conductance pathways, rather than interfering channels, dominate bullvalene constitutional effects.”

b. Line 189, a paragraph break at “In Fig. 4c” would help make the “In contrast” in p. 196 clear. As written, it is very confusing.

Response: the paragraph break has been entered and the next sentence rewritten.

“In Fig. 4c, an individual conductance trace is shown that is representative of 10% of the total number observed. This trace starts at the background level, jumps to ca. $8 \mu G_0$, jumps again to ca. $150 \mu G_0$, and then falls to the background level. The middle jump is indicative of junction isomerisation, whereas the terminal changes are associated with junction formation and breakage. The switching rate between $8 \mu G_0$ and $150 \mu G_0$ occurs at a rate of 2 h^{-1} with an average duration of 0.7 s for the blinks at the $150 \mu G_0$ level.”

c. Line 194 “is mostly not associated with continuous conduction at essentially the one level”. I do not understand the sentence or how it reflects the observations.

Response: this sentence was a Segway into the next paragraph and has now been deleted.

d. Line 297 “MD simulations of alkanedithiol junctions perceive”. Perception in its standard use refers to the action of understanding or realizing. I really cannot understand how MD simulations can “perceive”.

Response: we agree with the Reviewer. The term “perceived” has been changed to “indicate” in the revised manuscript.

“MD simulations of alkanedithiol junctions⁴⁸ indicate that molecules can bind to gold.”

e. Line 252, “the observed conductance traces are measured sampling each...” Here the word measured is confusing because it is not helping distinguish the experiments from the computations.

Response: We thank the Reviewer. This has been rewritten.

“Finally, conductance histograms are simulated and shown in Fig. 6f. These are obtained by directly mimicking the experimental procedure in which conductance is sampled at 0.033 ms intervals as the tip is retracted. KMC calculations are performed that simulate 3000 individual conductance traces and then average their properties. Each trace starts with the A_{mm} conformer at $d = 9.2 \text{ \AA}$, using a time step of 11 ns. At each time step, the reaction rates are determined

(Fig. 6c) and a new isomer assigned using a random number generator weighted by the relative rates. Using Fig. 6e, the conductance is then determined and averaged over the 3000 time steps comprising each 0.033 ms sampling interval. In this way, single-molecule conductance is attributed to *mixtures* of isomers that are in equilibrium on the timescale of the sampling period, with equilibration processes requiring longer timescales being excluded. As calculated isomeric conductances vary by a factor of 1000 (Fig. 6e), in general, small amounts of highly conducting isomers present in the equilibrium mixtures dominate the calculated conductance. Note that the mole fractions of each isomer at each d may be calculated by averaging over the 3000 trajectories (SI Fig. S6), giving results that agree with the direct kinetics simulations already reported in Fig. 6d.”

f. The whole paragraph at Line 268-278 is highly convoluted, making it challenging to extract its key scientific content.

Response: this paragraph has been rewritten as per our reply to comment (e) owing to changes in the figures introduced to meet subsequent comments.

g. The paragraph in Line 289 contains many speculative aspects that do not contribute to the clarity in presentations. For example, how are reaction rates on the 100 ns timescale in Line 295 are expected? What is the evidence for molecular binding prevent tip restructuring in Line 300. How do you define “success” for the 4-atom tip model in line 301. These are just some examples, of timescale and energies that are just not justified in the paper.

Response: we thank the Reviewer for their suggestion. The reaction rates come from transition-state theory. This has been now added to the text.

The original line 300 is a postulate that directly flows from the calculation results described in the previous line. It is this postulate that is to be tested by the subsequent calculations. Basically, this section highlights two decades of, what we believe, are internally inconsistent applications of computational methods to understand junction conductance.

An assortment of references has been added following the word “success”.

“MD simulations of alkanedithiol junctions⁴⁸ indicate that molecules can bind to gold even before the gold-gold contacts break, with the malleable molecules like alkanes becoming largely extended before breakage so that junctions do not form initially with low-conductance gauche conformers^{48,49}. Hence molecular binding could indeed inhibit tip restructuring, leading to the wide-ranging success^{15,16,51,52} of 4-atom-tip models for the interpretation of STMBJ data.”

h. It is unclear what is the evidence for the conclusion in line 364-367.

Response: in the mentioned line, we challenge the tipped models that have dominated calculations for so long. There is no support for tipped models except that quite a range of phenomena can be described that way. These phenomena turn out to be independent of detailed

assumptions as to the tip structure, and hence they work. The calculations performed in this work could have been done by many researchers over the past 2 decades, whereas they have been rarely performed. Testing these assumptions, as done in this work (and those that have looked at this before), show large effects. Clearly the results obtained apply to many calculations done in the past, certainly any DFT calculation done using the same level of theory (PBE). Nevertheless, as the effects have strong chemical origin and are large, it is reasonable to assume that any computational method capable of examining the question would yield similar results.

i. Some of the figures are unnecessarily complex. In particular, this reviewer highly encourages the authors to revisit the Fig. 5 and 6 as they are really hard to read. For example, wouldn't a figure that just focuses on the key species be easier to understand and allow the authors to communicate the chemistry at play? Or, wouldn't sublabel in the different plots help the reader read through the figure?

Response: we agree with the Reviewer and in the revised manuscript we have simplified and better labelled and improved all Figs. Both Figs. (5 & 6) have now been split into two Figs., with some material going into SI from each. This required large-scale rearrangements of the associated text.

The original Fig. 5a now appears as Fig. 5 (atomic structure), whilst 5b,c,d,e,f,h now appear in Fig. 6, and Fig. 5g is moved to SI as Fig. S15.

The revised figures include many of the figures of the previous version but with simplification as recommended by the Reviewer. We believe including detailed experimental and computational results are necessary in terms of understanding the features controlling conductance for the diaryl bullvalenes. To understand how much of each isomer is present on average, or in a simulated trace, requires knowledge of their relative energies and the energies of their interconverting transition states. This data then leads to rate constants for forward and reverse reactions, which leads to isomeric composition data. To get junction conductance, the conductances of each isomer then need to be determined. This is the logic of (newly numbered) Fig 6.

The top line of the original Fig. 6 is now expanded to include one more critical structure and is renumbered as Fig. 7, whilst the bottom line is moved to SI Fig S16, and the conductance simulations by MD are put into a new figure, Fig. 8. These results present a much more comprehensive model for understanding junction structure than widely-used approaches, showing that the 4-atom tip model is not tenable when bullvalenes bind to gold.

These figures and the associated text refer in detail to the names given to the constitutional and conformational isomers. This naming was originally done at the start of the results section under the heading experimental characterisation. It has been moved to a new results subsection preceding the subsections pertaining to these figures. As per the Reviewer's recommendation, this may help with reader comprehensibility, as now like topics are moved closer together.

In Fig. 5d it is unclear why the authors decided to use "speed up" instead of actual rates. While the modulation of chemical reactivity by nanoscale environments is of interest, it is not the focus of the discussion where actual rates instead of speed ups is what matters.

Response: the speedup factor is a property of critical interest as it tells how confinement in the junction controls reactivity, as described in the text. It is also important from a technical perspective in that calculations can obtain this quantity with some accuracy, whereas they are poor at calculating absolute rates. By focusing on the speedup factor, we can make realistic calculations expected to mimic experiment. The realistic prediction for the rate of the observed oscillating reaction is an example of the benefits of this approach. Concerning this, there is one place in the text where the absolute rate is the topic of discussion, and this sentence has now been augmented to remind readers how to convert the data in the figure into absolute rates.

The difference between f and g is also hard to understand from the figure itself.

Response: we agree and thank the Reviewer. Figure g has been now moved to SI. It contained data for isomers that was not critical in understanding the observed processes, with the most important information given initially in figure f.

j. Fig. 3 and 4 would really benefit from direct assignments of the different features observed that can be easily understood by just looking at the figures. Similarly, I don't think Fig. 1 does a good job at communicating the main physical picture that is emerging. There are many more examples of this type that, overall, make the paper confusing. In future revisions, the authors should carefully revisit their structure and writing to maximize clarity, fully justify all statements and claims and make their contributions and arguments completely transparent.

Response: all figures of the revised manuscript has been now labelled, simplified and in some instances split to impose more clarity. In Fig. 3, each panel has been now labelled such that it can be read on its own without referring to the text or caption, as suggested by the Reviewer. Same thing was performed for Fig. 4 including changing the y axis in (a), (c) and (d) from current to conductance to make it consistent with the text. This simplification of Fig. 4 was also requested by Reviewer 3.

Fig. 1 has also been changed to make the main ideas clearer, which are about bullvalene chemistry, junction binding, *in-situ* isomerisation measured in real time, and piezoresistance.

ii. While the computations presented represent a solid effort to understand the complexity of the experiment, the picture that emerges from these computations as summarized in Figures 5 and 6 is not clear and, further, it disagrees with experiment. Thus, at the end of the day the computations are not offering a clear atomistic picture that can be used to understand the experiments. It would be advisable for the authors to decide what are reliable computational results that can be used in this paper, and what part of the computations are still at a level in which the results depend strongly on "ad hoc" undefined parameters and thus not publishable.

Response: we thank and respect the Reviewer's suggestion. When calculations are performed on such complicated systems as molecular junctions, perfect agreement with experiment is not

to be expected. All calculations, including these suffer from the problem that junction structures remain unknown at the atomic level, and that realistic calculations to predict structures are a long way from being computationally feasible. Hence, some model must always be assumed, with assumptions made in setting up the calculations being described here as “ad hoc”. All calculations performed over the last two decades for single-molecule junctions make such “ad hoc” assumptions. What is different about the current paper is that unjustified assumptions appearing in most previous works are carefully examined, leading to a more general approach taken that could produce the common models as its output, or else generate something completely different.

In addition, all calculations make approximations as to the level of accuracy to which forces and structures are determined. The current calculations use PBE, which is sophisticated and expensive and generally applicable, yet its absolute accuracy is poor compared to that required to make quantitative chemical predictions. Hence, we believe that calculations need to report not only those aspects that agree with experiment but also those that do not. Indeed, some focus has been made in this paper concerning those that do not. We believe that this is to not only help with assessment of the main conclusions, but also to help future design of better methods.

To our knowledge, the kinetic Monte Carlo techniques developed herein to simulate conductance traces is unique. Never before have simulations attempted to calculate the details of the experimental setup and mimic its operation. This, to a large extent, removes assumptions that have dominated previous modelling in this field. It leads to the understanding of features of bullvalene isomerisation that could not be discovered by any previous simulations.

The primary elements of the tip restructuring in Figs. 7 and 8 are not new, but the application of such ideas in the field of molecular electronics has been rare. We believe that the conclusion reached concerning tip reconstructing surpasses previous knowledge and sets an agenda for future research. What we do not know is if this effect seen here is specific to bullvalenes owing to their shape-shifting properties, or if the result is widely general but only manifested easily owing to the observation of shape-shifting isomerisation for the bullvalenes. There is much yet to discover.

The paper is not written in chronological order, but instead as “experiments” followed by “interpretation” to clearly delineate what is established from what is surmised. The results section has four main parts:

- 1) STMBJ measurements and data analysis
- 2) Oscillating reaction measurements and data analysis
- 3) Calculations using a standard atomic-tip model
- 4) Calculations that predict violation of the standard tip model

In chronological order, the measurements in (1) and some of the data analysis was completed first, followed by (2) and, owing to its failure to describe key observed features, (3). Then new data analysis approaches to (1) were introduced and the experiments in (2) were designed, both to investigate properties predicted by the calculations. The results ranged from quantitative verification of the calculation results in (4) to the demonstration of features that sometimes agreed with calculations and sometimes did not in (1).

The purpose in doing the calculations was to not only provide interpretation but also to direct experiment and its analysis by highlighting features of interest. By doing calculations that are

constrained by some implicit or explicit action to give results expected to agree with experiment, all that is achieved is the demonstration that the constraints are reasonable for the interpretation of what is known. That is not the approach taken herein. Instead, calculations are made that are capable of giving wide ranging results, with no guarantee that they will match the experimental observations. We believe that this is how new phenomena can be discovered, and how experiments and their associated data analyses can be improved.

We made every effort in the revised manuscript to address the Reviewer's comment by simplifying the figures and the text and to move to the SI data that were less critical in understanding the observed processes.

Other points:

1. The authors argue that they have a piezoresistor that is stable for several Angstroms. But why is this unique? If I just pull an alkane I will get a piezoresistor and I can choose the length of stability of the piezoresistor simply by changing the length of the alkane. See, for instance:

<https://pubs.acs.org/doi/pdf/10.1021/acs.jpcllett.7b03323>

2. There are other contributions mechanically inducing changes in conductance that may be of interest to the authors.

a. Folding a single-molecule junction: <https://pubs.acs.org/doi/10.1021/acs.nanolett.0c02815>

b. Ring-opening reactions:

<https://pubs.rsc.org/en/content/articlelanding/2019/sc/c8sc04830d>

c. Association reactions:

<https://pubs.acs.org/doi/10.1021/acs.jpcc.1c01901>

d. Control of anchoring:

<https://journals.aps.org/prl/abstract/10.1103/PhysRevLett.121.047701>

e. Control of intermolecular interactions

<https://www.nature.com/articles/nchem.2588>

These papers provide clear examples where isomerism both constitutional and configurational can be used to control transport. In the context of this literature (and related papers) it would be useful for the authors to clearly delineate their advances. ‘

Response: we thank the Reviewer for their valuable suggestions. To address the Reviewers comment, we have now included references, a, b, c, and e into the general reference list concerning these topics in the introduction. Then these papers are also mentioned later in the text at directly relevant points. The inclusion of references a and e is significant as they discuss pi-stacking of aryl groups, a feature of this paper that previously was not explicitly mentioned.

Concerning the papers:

- a) Provides elegant experiments and calculations for a conformationally induced piezoresistive effect and so is directly relevant. It manifests 10-fold changes in resistivity, similar to what is observed here. The difference is that the bullvalenes offer pathways to much larger changes whereas (a) describes an effect that is limited in scope. Also, no oscillating reactions are observed in (a), nor evidence for multiple types of junctions. A conclusion from our paper is to challenge the validity of the junction model that is assumed and then used to interpret the data in (a).
- b) This is an important computational paper that combines empirical force field with semiempirical DFT to look at a chemical reaction in a junction and its effect on conductance. Reversibility of the reaction, and the possibility of seeing oscillating

reactions, is not considered. Empirical conductance calculations are employed. A tipped junction model is assumed in the calculations, the type of which our paper discuss alternatives to.

- c) This article investigates an interesting long graphene nanoribbons being formed on surfaces and pulled off to make long junctions. It involves experiment and simulation. There is no oscillating reaction, and no dependence of conductance on a controllable process other than what is expected based on McConnell's equation, as modulated by defects.
- d) This article involves experiments and calculations on piezoresistivity associated with pi-stacked dimers and is highly relevant. No oscillating reaction however is possible, and the paper does not discuss possible molecule-electrode contacts that our work discusses.

These papers all are significant and share some aspects with our work. However, none of them:

- Address oscillating reactions
- Provide a development pathway forward for improved piezoresistive devices
- Perform the extensive kinetic Monte Carlo analysis for dozens of isomers as a function of junction extension, nor perform simulations of individual traces to directly mimic experimental procedure
- Challenge long-used but unverified models for the nature of junction interface to gold electrodes and their effect on controlling qualitative conductance properties

For example, in our abstract:

“Constitutional and conformational isomerisation of diarybullvalens is exploited to demonstrate a single-molecule nano electro-mechanical system (NEMS) piezoresistor using the scanning tunnelling microscopy break junction (STMBJ) technique.”

-None of these papers considers both constitutional and conformational isomerism.

“Single-molecule constitutional isomerisation at 850 Hz is also observed using blinking (current-time) experiments performed at fixed junction separations.”

-None of these papers considers oscillating reactions.

“A kinetic Monte Carlo (KMC) methodology is developed to simulate isomerisation on the experimental timescale, parameterised using density-functional theory (DFT) combined with non-equilibrium Green's function (NEGF) calculations.”

-None of these papers develops methods that can handle dozens of isomers, and none of the papers directly simulates observed conductance traces.

“Results indicate that piezoresistance is controlled by both constitutional and conformational isomerisation occurring at rates that are either fast (equilibrium) or slow (non-equilibrium) compared to the experimental timescale.”

-None of these papers discuss extensive competing chemical processes that can be controlled to be faster or slower than the timescale of the experiment.

“Two different types of STMBJ traces are observed, one typical of traditional experiments that is interpreted in terms of isomerisation occurring within tipped-junction atomic models, and

another attributed using DFT/NEGF molecular dynamics (MD) simulations to arise from junction-interface restructuring induced by bullvalene isomerisation.”

-None of these papers observes or predicts such differences in conductance properties, and none of these papers challenges the basic assumptions that have been used for decades concerning gold–electrode junction structure.

Hence, the Abstract describes features that are not included in the papers listed. The Conclusions section also addresses each point from the Abstract. Of course the listed papers are important and relevant, as indeed are the many other papers mentioned and/or discussed.

3. The authors use “conductivity” and “piezoresistivity”. However, these quantities refer to intensive quantities that are ill-defined at the single-molecule limit. The correct terms, I would say, are “conductance” and “piezoresistance”.

Response: we thank and agree with the Reviewer. These have been changed (except in titles in the references).

Reviewer #3 (Remarks to the Author):

I enjoyed reading this manuscript. The choice of molecules for study and the design of the experiments/calculations were very clever. The experiments and calculations appear to have been done thoroughly and competently. The results are new and will be of interest to a wide community of researchers. I recommend that the manuscript be published without significant modifications.

Response: we thank the Reviewer for their positive evaluation.

Several small points:

Page 2, paragraph 3: “First, piezoresistivity in demonstrated that is stable...” should be “First, piezoresistivity is demonstrated that is stable”.

Response: we thank the Reviewer for their careful reading. This has now been corrected.

Page 3, paragraph 3: “...of which B, D, and E each form sets of enantiomeric stereoisomers...” In the caption of Figure 2, the enantiomeric pairs are noted as “(A, A'), (B, B'), and (D, D')” which is correct?

Response: we again thank the Reviewer for their careful reading. This has now been corrected.

Page 4, paragraph 2: The authors mention “a non-isomerisable control molecule”. It appears from the SI that this is “para - (4,4'-bi(methylthio)terphenyl)”. I suggest the authors simply name the control molecule in the text. I don't believe there is a reason not to. This non-isomerisable molecule is mentioned again on p. 5.

Response: this has now has been corrected.

Page 4, paragraph 2: “Figure 3b shows that these conductance peaks arise as the tip is retracted to separations too large to support conductance via the mechanism operative for the large peak (red arrow).” The authors' Figures are, of necessity, rather complicated, and each contains a lot of information. As I was reading the text and referring back to the Figures, I started looking for a red arrow in Figure 3b, only to realize (after an embarrassingly long search) the red arrow referred to was the red arrow from Figure 3c. I suggest helping future readers: “Figure 3b shows that these conductance peaks arise as the tip is retracted to separations too large to support conductance via the mechanism operative for the large peak (red arrow in Figure 3c).” I admit that this is a very minor point.

Response: we thank the Reviewer. The sentence has been rewritten for clarity.

Concerning Figure 3b: There are two faint circles down on the plot, but I could find no reference to them in either the caption or the text. It is easy for the reader to imagine the meaning of these circles, but the authors may want to add a sentence to make the meaning unambiguous.

Response: the caption to Fig 3b has been expanded to describe the circles.

“Circles highlight the features that correspond to the those highlighted by arrows in Fig. 3c.”

Page 5, paragraph 2: “In Fig. 4c, an individual conductance trace, representative of 10% of the total number observed, is shown a current trace that starts at the background level, jumps to ca. $8 \mu G_0$, jumps again to ca. $150 \mu G_0$,” The text uses units of μG_0 , but the figure uses nA. Wouldn't it be cleaner to use μG_0 throughout?

Response: we thank and agree with the Reviewer. The y axis (previously current) in Fig. 4 of the revised manuscript has been changed to μG_0 . We previously plotted the blinks y axis as current as it enables showing the magnitude of the tunnelling current. We then made the conversion to conductance in the histograms only. As per the Reviewer's suggestion, in the revised manuscript we now plot the y axis as conductance. We also added in the caption that to calculate the conductance, one need to subtract the tunnelling conductance from the blinking conductance.

Fig. 4 | Blinking current–time traces (a) representative blinks at specific separation between the tip and the surface at 11.8 Å (red trace), 12.8 Å (brown trace) and 16.2 Å (blue trace). The surface bias was +100 mV. (b) Conductance histograms built from 100 blinks for each distance; arrows at low conductance indicate the background tunnelling between the two gold electrodes in the absence of a molecule, whereas arrows at high conductance indicate through-molecule conductance. (c) An example of a blink, which represent 10% of the blinks observed at 16.2 Å, that switches from ca. $10 \mu G_0$ to $100 \mu G_0$. **The molecular conductance is the difference between the blinking conductance and the background tunnelling conductance.** (d) Typical of 80% of blinks at 16.2 Å, showing the current fluctuating by 2–5 fold at 750–1000 Hz, unlike the blinks observed at 11.8 and 12.8 Å. (e) FFT analysis of the blinks observed at 16.2 Å assigned to bullvalene switching. (f) FFT analysis of the background tunnelling current in the absence of a connecting molecule.

Page 7, paragraph 3: Reference to Figure 3c should be to Figure 4c.

Response: we again thank the Reviewer for their careful reading. This has now been corrected.

I congratulate the authors on a splendid piece of science!

REVIEWERS' COMMENTS

Reviewer #1 (Remarks to the Author):

Since the authors answered my comments satisfactorily, I recommend its publication.

Reviewer #2 (Remarks to the Author):

Review of the revised version of "Controlling Single-Molecule Piezoresistivity through Isomerisation of Bullvalenes" by Reimers et al.

I have read carefully the revised version of this paper, which represents a significant improvement over the original submission. Many of the concerns that I raised have been appropriately addressed by the authors.

A few more comments:

1. In page 8, I would recommend removing the sentences between "By increasing the activation energy... unlikely." The validity of the approach in the SI that leads to this sentence is difficult to rigorously justify and, for this reason, this reviewer cannot be supportive of highlighting it in the main text.

2. Significant effort was made by the authors to clean up their discussion of the extensive computational results. The results do not agree with experiments and the authors have argued that they are embracing and emphasizing the disagreement. While it is true that in molecular electronics quantitative simulations are challenging to obtain due to the uncertainty in junction configuration, correct qualitative trends are often extracted.

Right now, the paper has a very interesting model for junction formation and evolution based on KMC that I believe has all the right elements to yield correct interpretation of the experiments, and to describe the conductance histograms. The authors suggest that they don't achieve this because their model does not include tip reconstruction and provide computations that are suggestive that this interpretation is plausible.

I find this last component of the paper to be underdeveloped. In principle, the way forward is clear. If the interpretation of the tip reconstruction is correct, then doing the KMC with structures with 4-atom Au tips and collapsed tips should offer semi-quantitative agreement with experiments. Right now, the computational efforts in the paper, while clearly extensive, are only half way to match the experiments.

This reviewer realizes that this new set of computations represent a significant computational effort. Thus, leaves it as optional, but strongly encouraged, for the authors to consider completing the simulation effort and establish the KMC approach through this contribution as a method to capture the conductance histograms, and the full complexity and interconversion in molecular electronics!

3. There are a few “conductivities”/“piezoresistivity” remaining in the text. This include:

- a. Page 9, paragraph before Conclusions
- b. Fig. 1d and also in the caption.

Typos:

1. Fig. 3d. “Conductance” in the labels
2. Table S5, “Fig. 8e” should be “Fig. 7e” I believe.

Reviewer #3 (Remarks to the Author):

The authors have dealt with my comments quite well. I believe their manuscript should be published as it stands.

REVIEWERS' COMMENTS

Reviewer #2 (Remarks to the Author):

Review of the revised version of “Controlling Single-Molecule Piezoresistivity through Isomerisation of Bullvalenes” by Reimers et al.

I have read carefully the revised version of this paper, which represents a significant improvement over the original submission. Many of the concerns that I raised have been appropriately addressed by the authors.

Response: we thank the Reviewer for their positive feedback.

A few more comments:

1. In page 8, I would recommend removing the sentences between “By increasing the activation energy... unlikely.” The validity of the approach in the SI that leads to this sentence is difficult to rigorously justify and, for this reason, this reviewer cannot be supportive of highlighting it in the main text.

Response: we thank the Reviewer for their suggestion. The justification for this comes from the Arrhenius equation (which is now named so on SI page 7). Small errors in the calculations could lead to qualitatively incorrect predictions for properties such as the produced ratios of alternate reaction products. Hence, we feel that discussion of this issue is required in the main text. To address the Reviewers' comment, the main text has now been slightly modified to quote some of the key results presented in the SI, which indicate the high sensitivity of the conductance-trace calculations to small changes in the reaction energetics. A key point made previously in both the SI and main text is that the conceptual design of the calculations is to minimise the impact of such computational errors in the reaction energetics by calculating differential properties and then comparing to differential observed properties. Text was modified p. 8: By increasing the activation energy for **A** to **D** conversion by 0.10 – 0.13 eV, the third scheme predicts that 70 % – 90% of conductance traces should dissociate before isomerisation.

2. Significant effort was made by the authors to clean up their discussion of the extensive computational results. The results do not agree with experiments and the authors have argued that they are embracing and emphasizing the disagreement. While it is true that in molecular electronics quantitative simulations are challenging to obtain due to the uncertainty in junction configuration, correct qualitative trends are often extracted.

Right now, the paper has a very interesting model for junction formation and evolution based on KMC that I believe has all the right elements to yield correct interpretation of the experiments, and to describe the conductance histograms. The authors suggest that they don't achieve this because their model does not include tip reconstruction and provide computations that are suggestive that this interpretation is plausible.

I find this last component of the paper to be underdeveloped. In principle, the way forward is clear. If the interpretation of the tip reconstruction is correct, then doing the KMC with

structures with 4-atom Au tips and collapsed tips should offer semi-quantitative agreement with experiments. Right now, the computational efforts in the paper, while clearly extensive, are only half way to match the experiments.

This reviewer realizes that this new set of computations represent a significant computational effort. Thus, leaves it as optional, but strongly encouraged, for the authors to consider completing the simulation effort and establish the KMC approach through this contribution as a method to capture the conductance histograms, and the full complexity and interconversion in molecular electronics!

Response: we agree with the Reviewer that the simulations presented do not fully provide a thorough understanding of the chemical processes that take place during a break junction experiment. Yet the KMC approach goes far beyond any previous modelling of conductance traces, and the subsequent dynamics approach is analogous to the best works ever performed to get at what real junctions could look like.

As acknowledged by the Reviewer, calculations that would make a major advance on what is presented herein would take full atomic models of the tip and surface in solution to directly model the crash and then subsequent extraction, following the flow of the active molecule into the developing gap and its surface interactions. The dynamics would need to be solved using accurate forces pertaining to a huge range of chemical environments, by applied to huge systems, and taken out from the intrinsic femtosecond electronic timescale to the microsecond tip-retraction timescale. The computational cost for doing so is beyond what can be envisaged in the foreseen future.

3. There are a few “conductivities”/”piezoresistivity” remaining in the text. This include:
 - a. Page 9, paragraph before Conclusions
 - b. Fig. 1d and also in the caption.

Response: these typos have been changed now in the revised manuscript.

Typos:

1. Fig. 3d. “Conductance” in the labels

Response: we thank the Reviewer. These typos have been changed now in the revised manuscript.

2. Table S5, “Fig. 8e” should be “Fig. 7e” I believe.

Response: we thank the Reviewer. This has been fixed now.